# Advantage Shaping as Surrogate Reward Maximization: Unifying Pass@K Policy Gradients

**Christos Thrampoulidis**          *cthrampo@ece.ubc.ca*
**Sadegh Mahdavi**                  *smahdavi@ece.ubc.ca*
**Wenlong Deng**                    *dwenlong@ece.ubc.ca*
*Department of Electrical and Computer Engineering*
*University of British Columbia*

**Reviewed on OpenReview:** *https://openreview.net/forum?id=R1RhBFUk8t*

## Abstract

We unify two seemingly distinct approaches to policy gradient optimization for the Pass@K objective in reinforcement learning with verifiable rewards (RLVR): direct REINFORCE-style methods and advantage-shaping techniques that modify GRPO. By reverse-engineering existing advantage-shaping algorithms, we reveal that they implicitly optimize surrogate rewards. We specifically interpret practical "hard-example upweighting" modifications to GRPO as reward-level regularization. Conversely, starting from surrogate reward objectives, we provide a simple recipe for deriving both existing and new advantage-shaping methods. This perspective provides a lens for RLVR beyond our original motivation of Pass@K.

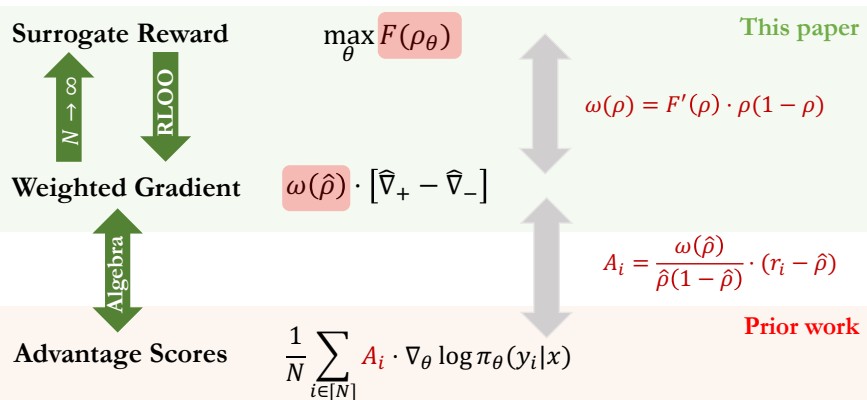

Figure 1: Illustration of our bidirectional framework mapping Advantage Scores ($A_i$) of RLVR algorithms with 0/1 and Pass@K rewards, such as GRPO Shao et al. (2024), Pass@K-GRPO Chen et al. (2025); Mahdavi et al. (2025), and Prioritized Sampling Team et al. (2025), to Surrogate Rewards ($F(\rho)$) via Gradient Reweighting ($\omega(\hat{\rho})$) of the average gradient updates of correct ($\widehat{\nabla}_+$) and wrong ($\widehat{\nabla}_-$) model responses. The framework provides a recipe for engineering new RLVR algorithms by choosing a surrogate reward objective (or equivalently, a reweighting function).

## 1 Introduction

When evaluating large language models on math and coding tasks, standard practice is to generate $K$ independent solutions and check whether any succeeds; the so-called Pass@K metric (Kulal et al., 2019; Chen et al., 2021). Yet most policy gradient methods, from REINFORCE (Williams, 1992) to more recent variants like RLOO (Kool et al., 2019; Ahmadian et al., 2024) and GRPO (Shao et al., 2024), optimize for single-attempt reward creating a mismatch. Should we instead train what we test?

Recent work has approached this question from two seemingly distinct angles. Mahdavi et al. (2025); Walder & Karkhanis (2025) derive policy gradients that directly maximize Pass@K reward, yielding REINFORCE-style updates with reweighted advantages. Meanwhile, Chen et al. (2025) propose modifying GRPO through *advantage shaping*, that is, directly adjusting advantage scores to account for the Pass@K objective. Both improve Pass@K performance, but their relationship remains unclear. *Are these fundamentally different techniques, or two views of the same underlying principle?*

This work unifies these seemingly distinct perspectives, showing that advantage shaping and direct optimization are two sides of the same coin. The key is recognizing that different policy gradient methods can be understood as optimizing different *surrogate rewards*; see Figure 1 and Table 1. More broadly, we establish surrogate rewards as a design mechanism for policy gradient optimization with verifiable rewards, providing a recipe for algorithm interpretation via reverse-engineering and development via forward-engineering.

## 1.1 Problem Setup: Reinforcement Learning with Verifiable Rewards

We study reinforcement learning with verifiable rewards (RLVR) (Lambert et al., 2024; Shao et al., 2024; Guo et al., 2025), a framework for training Large Language Models (LLMs) on tasks with objectively verifiable outcomes (see (Zhang et al., 2025) for a recent survey). Concretely, we are given a distribution $\mathcal{P}$ over problem-answer pairs $(x, a)$, where $x$ might be a math problem and $a$ its final numeric answer. The goal is to train an LLM to maximize the expected reward of finding correct answers to problems from $\mathcal{P}$:

$$\max_\theta \quad \mathbb{E}_{(x,a)\sim\mathcal{P}}\left[\mathbb{E}_{y\sim\pi_\theta(\cdot|x)}\, r(y,a)\right]. \tag{1}$$

We abstract the LLM as a conditional distribution $\pi_\theta(\text{response}|\text{prompt})$ parameterized by $\theta$: Given prompt $x$, the model generates response $y$ (a sequence of tokens) by autoregressive sampling. The reward function $r(y, a)$ assesses whether the model response $y$ agrees with the reference answer $a$. If it does, the response is *correct*; otherwise it is *wrong*. RLVR assumes the reward can be verified externally. Note the nested expectation: first over problem-answer pairs $(x, a) \sim \mathcal{P}$, then over model responses $y \sim \pi_\theta(\cdot|x)$ per problem.

### 1.1.1 The 0/1 Reward

The most popular reward function in RLVR is the binary 0/1 reward:

$$r_{0/1}(y,a) := \begin{cases} 1 & \text{if } y \text{ agrees with } a \\ 0 & \text{otherwise} \end{cases}.$$

Model performance is measured by the expected 0/1 reward over all examples $(x, a)$ in the distribution: $\mathbb{E}_{(x,a)\sim\mathcal{P}}[\rho_\theta(x,a)]$, where the per-example expected 0/1 reward (equal to the probability that model $\theta$ generates a correct response for problem $x$) is:

$$\rho_\theta(x,a) := \mathbb{E}_{y\sim\pi_\theta(\cdot|x)}\, r_{0/1}(y,a).$$

A popular algorithm for maximizing expected 0/1 reward is REINFORCE Leave-One-Out (RLOO) (Kool et al., 2019; Ahmadian et al., 2024), which extends the naive REINFORCE algorithm (Williams, 1992) using multiple generated responses to form a baseline. At each iteration $t$, and for each pair $(x, a)$ from a training set $\mathcal{D}$, RLOO updates the model parameters as

$$\theta_{t+1} = \theta_t + \eta \cdot \widehat{G}_{\text{RLOO}}(\theta_t; (x,a)),$$

where, $\eta$ is the learning rate, and the gradient estimate is a weighted average over $N$ generated responses $y_i \overset{\text{IID}}{\sim} \pi_\theta(\cdot|x)$:

$$\widehat{G}_{\text{RLOO}}(\theta; (x,a)) = \frac{1}{N}\sum_{i\in[N]} A_i^{\text{RLOO}} \cdot \nabla_\theta \log \pi_\theta(y_i|x). \tag{RLOO}$$

The weights $A_i^{\mathrm{RLOO}}$ are called *advantage scores.* The construction of RLOO's advantages scores ensures $\widehat{G}_{\mathrm{RLOO}}$ is an unbiased, low-variance estimate of $\nabla_\theta \rho_\theta(x, a)$, the gradient of the population (expected) 0/1 reward. Intuitively, for large $N$:

$$\widehat{G}_{\mathrm{RLOO}}(\theta; (x, a)) \approx \nabla_\theta \rho_\theta(x, a). \qquad (2)$$

Despite its simplicity, RLOO is a strong baseline (Ahmadian et al., 2024). Its structure also underlies recent popular variants like Group Relative Policy Optimization (GRPO) (Shao et al., 2024). In full online mode (when clipping can be ignored), GRPO uses the same form as (RLOO) with normalized advantages $A_i^{\mathrm{GRPO}} = A_i^{\mathrm{RLOO}}/\sqrt{\hat{\rho}(1 - \hat{\rho})}$, where $\hat{\rho}$ is the *empirical* 0/1 reward for example $(x, a)$.

### 1.1.2 The Pass@K Reward

The 0/1 reward, however, misaligns with practical LLM usage. Given a problem $x$, users typically prompt the model multiple times (say, $K$ times) and examine the responses $y_1, \ldots, y_K$ to find one that produces the desired answer. Accordingly, LLMs are often evaluated using the Pass@K metric (Chen et al., 2021), which measures whether at least one solution is correct among $K$ generated responses:

$$r_{\mathrm{Pass@K}}(\{y_i\}_{i \in [K]}, a) = \begin{cases} 1 & \text{if at least one } y_i \text{ agrees with } a \\ 0 & \text{otherwise} \end{cases}.$$

Since test-time performance is measured via Pass@K, we may train the model to directly maximize Pass@K reward rather than 0/1 reward by maximizing $\mathbb{E}_{(x,a) \sim \mathcal{P}}[\rho_{K,\theta}(x, a)]$, where we define the per-example expected Pass@K reward as

$$\rho_{K,\theta}(x, a) := \mathbb{E}_{\{y_i\}_{i \in [K]} \sim \pi_\theta(\cdot|x)} \, r_{\mathrm{Pass@K}}(\{y_i\}_{i \in [K]}, a).$$

This equals the probability that model $\theta$ generates at least one correct response in $K$ samples.

### 1.2 Contributions

**C-1. Direct Pass@K optimization.** Building on Mahdavi et al. (2025), we show that the exact Pass@K policy gradient equals the 0/1 gradient reweighted by per-example Fail@(K-1) probabilities. This yields Monte Carlo unbiased estimators $\mathrm{REINFORCE}_K/\mathrm{RLOO}_K$ and a biased variant $\mathrm{GRPO}_K$, all expressible as *asymmetric* reweightings of their 0/1-reward counterparts.

**C-2. Advantage shaping = surrogate reward maximization.** We reverse-engineer the advantage-shaping algorithm of Chen et al. (2025) and find that it asymptotically (in the number of generated responses) maximizes a surrogate reward $\frac{2}{K} \arcsin(\sqrt{\rho_{K,\theta}})$.

Conversely, we provide a forward-engineering recipe: starting from any differentiable surrogate $F : [0, 1] \to \mathbb{R}$ and surrogate-reward maximization objective

$$\max_\theta \ \widehat{\mathbb{E}}_{(x,a) \sim \mathcal{D}} \left[ F \left( \mathbb{E}_{y \sim \pi_\theta(\cdot|x)} \, r(y, a) \right) \right], \qquad (3)$$

advantage-shaping heuristics can be systematically derived as follows:

    (i) Differentiate the surrogate reward: $\nabla_\theta F(\rho_\theta) = F'(\rho_\theta) \cdot \nabla_\theta \rho_\theta$
    (ii) Substitute all population quantities with their empirical analogues:
        (a) Replace $F'(\rho_\theta)$ with $F'(\hat{\rho})$
        (b) Replace the population reward gradient $\nabla_\theta \rho_\theta$ with its RLOO proxy $\widehat{G}_{\mathrm{RLOO}}(\theta)$ (Eq. (2)).

Put together, this yields a per-example empirical gradient $\widehat{G}_F(\theta; (x, a)) = \frac{1}{N} \sum_{i \in [N]} A_i^F \cdot \nabla_i$ with advantage shaped as $A_i^F = F'(\hat{\rho}) \cdot \sqrt{\hat{\rho}(1 - \hat{\rho})} \cdot A_i^{\mathrm{GRPO}}$. Thus, a practical implementation can reuse existing GRPO code by multiplying the GRPO advantages (before clipping) by this prefactor.

As a corollary, when $K{=}1$, we show GRPO is (up to clipping) effectively RLOO applied to the surrogate reward $F(\rho_\theta) = 2 \arcsin(\sqrt{\rho_\theta})$, a variance-stabilizing transformation of the 0/1 reward.

**C-3. Reward-level regularization.** We interpret commonly used heuristics that downweight "easy" examples (those already solved with high probability) and upweight "hard" ones (Chen et al., 2025; Team et al., 2025). Concretely, we show that multiplying the GRPO gradients by $1 - \widehat{\rho}$ is effectively equivalent to optimizing a regularized surrogate reward $\arcsin(\sqrt{\rho_\theta}) + \sqrt{\rho_\theta(1 - \rho_\theta)}$. More generally, this can be interpreted as a special instance of regularized surrogate reward maximization:

$$\max_\theta \ \widehat{\mathbb{E}}_{(x,a)\sim\mathcal{D}} \left[ F\left( \mathbb{E}_{y\sim\pi_\theta(\cdot|x)}\, r(y,a) \right) + \lambda \cdot \Omega\left( \mathbb{E}_{y\sim\pi_\theta(\cdot|x)}\, r(y,a) \right) \right], \tag{4}$$

where the stochastic training objective over examples $(x,a) \sim \mathcal{D}$ consists of two components: a data-fitting term $F$ (e.g., the GRPO $\arcsin(\sqrt{\rho_\theta})$ surrogate) and an additive regularizer $\Omega$ that encourages maintaining some probability mass on wrong responses, thereby exploring alternative solution paths that may generalize beyond the training set. Unlike typical policy-level or parameter-level regularization, this form of regularization operates simply at the reward level. To demonstrate its use, we instantiate the recipe of Contribution C-2 with entropy regularization, yielding a new tunable advantage-shaping algorithm.

Table 1: Unified view of main algorithms studied in this work. Rows grouped by target evaluation metric. $\rho, \rho_K$ are per-example expected 0/1 and Pass@K rewards; $\hat{\rho}, \hat{\rho}_K$ are their empirical estimators.

| Algorithm | Advantage Scores $(A^+, A^-)$ | Weighted Empirical Gradient | Population Surrogate Reward |
|---|---|---|---|
| *Targets 0/1 evaluation (K=1)* | | | |
| RLOO (Kool et al., 2019) | $\left( \frac{N(1-\hat{\rho})}{N-1},\ -\frac{N\hat{\rho}}{N-1} \right)$ | $\hat{\rho}(1-\hat{\rho})\,[\widehat{\nabla}_+ - \widehat{\nabla}_-]$ | $\rho$ |
| GRPO (Shao et al., 2024) | $\left( \sqrt{\frac{1-\hat{\rho}}{\hat{\rho}}},\ -\sqrt{\frac{\hat{\rho}}{1-\hat{\rho}}} \right)$ | $\sqrt{\hat{\rho}(1-\hat{\rho})}\,[\widehat{\nabla}_+ - \widehat{\nabla}_-]$ | $2\arcsin(\sqrt{\rho})$ |
| Skew-R [This work][1] | $\left( (1-\hat{\rho})\sqrt{\frac{1-\hat{\rho}}{\hat{\rho}}},\ -(1-\hat{\rho})\sqrt{\frac{\hat{\rho}}{1-\hat{\rho}}} \right)$ | $(1-\hat{\rho})\sqrt{\hat{\rho}(1-\hat{\rho})}\,[\widehat{\nabla}_+ - \widehat{\nabla}_-]$ | $\arcsin(\sqrt{\rho}) + \sqrt{\rho(1-\rho)}$ |
| *Targets Pass@K evaluation (K≥2)* | | | |
| RLOO$_K$ [This work] | $\left( \widehat{f}_{K-1}^+ \frac{N(1-\hat{\rho})}{N-1},\ -\widehat{f}_{K-1}^- \frac{N\hat{\rho}}{N-1} \right)^2$ | $\widehat{f}_{K-1}^+\,\hat{\rho}(1-\hat{\rho})\,\widehat{\nabla}_+ \ - \ \widehat{f}_{K-1}^-\,\hat{\rho}(1-\hat{\rho})\,\widehat{\nabla}_-$ | $\rho_K$ |
| $\widetilde{\mathrm{GRPO}}_K$ (Chen et al., 2025) | $\left( \widetilde{\omega}_K \sqrt{\frac{1-\hat{\rho}}{\hat{\rho}}},\ -\widetilde{\omega}_K \sqrt{\frac{\hat{\rho}}{1-\hat{\rho}}} \right)^3$ | $\sqrt{\frac{1-\hat{\rho}_K}{\hat{\rho}_K}}\,\hat{\rho}\,[\widehat{\nabla}_+ - \widehat{\nabla}_-]$ | $\frac{2}{K}\arcsin(\sqrt{\rho_K})$ |
| GRPO$_K$ [This work] | $\left( \widehat{f}_{K-1}^+ \sqrt{\frac{1-\hat{\rho}}{\hat{\rho}}},\ -\widehat{f}_{K-1}^- \sqrt{\frac{\hat{\rho}}{1-\hat{\rho}}} \right)$ | $\widehat{f}_{K-1}^+ \sqrt{\hat{\rho}(1-\hat{\rho})}\,\widehat{\nabla}_+ \ - \ \widehat{f}_{K-1}^- \sqrt{\hat{\rho}(1-\hat{\rho})}\,\widehat{\nabla}_-$ | $\mathcal{B}\left( 1 - (1-\rho_K)^{1/K}; \frac{1}{2}, K - \frac{1}{2} \right)^4$ |

Table 1 unifies the main Pass@K policy-gradient algorithms studied in this work. Starting from their advantage scores, we express every method (middle column, ignoring normalization constants and clipping) as $w_+ \widehat{\nabla}_+ - w_- \widehat{\nabla}_-$, where effective gradient weights $w_\pm$ multiply the average empirical gradients over correct/wrong responses. Leveraging this unified representation, we reverse-engineer the corresponding population surrogate reward that each algorithm implicitly optimizes (last column). Conversely, using the forward-engineering recipe from Contribution C-2, we can map a chosen surrogate reward back to advantage scores, establishing a bidirectional relationship (see also Figure 1):

*advantage scores* $\longleftrightarrow$ *weighted gradients* $\longleftrightarrow$ *surrogate reward*

**Roadmap.** The rest of the paper is organized as follows. We start in Sec. 2 by reviewing the 0/1-reward setting and fixing the unified gradient/advantage notation that we use throughout. Sec. 3 then develops **C-1**: we directly differentiate the Pass@K objective leading to the REINFORCE$_K$/RLOO$_K$ estimators and the GRPO$_K$ variant. Sec. 4 presents the complementary *advantage-shaping* route of $\widetilde{\mathrm{GRPO}}_K$ by Chen et al. (2025) and contrasts its symmetric reweighting with the asymmetric reweighting arising from direct Pass@K optimization. Sec. 5 establishes **C-2** by bridging these views: we reverse-engineer $\widetilde{\mathrm{GRPO}}_K$ as (asymptotically) optimizing an arcsin-transformed surrogate Pass@K reward (Sec. 5.1), recover it by forward-engineering from this surrogate (Sec. 5.2), and then generalize the reverse/forward correspondence to a broad class of binary-reward RLVR gradients (Sec. 5.3). Finally, Sec. 6 develops **C-3** by interpreting practical hard-example upweighting as *reward-level regularization* and instantiating the forward-engineering recipe

---

[1] Effectively equivalent to Kimi 1.5 Prioritized Sampling (Team et al., 2025); see Sec. 6.2.
[2] $\widehat{f}_{K-1}^+, \widehat{f}_{K-1}^-$ are leave-one-out Fail@(K−1) scalers (Eq. (15)). Mahdavi et al. (2025) implements a biased version; App. C.3.
[3] The scaler $\widetilde{\omega}_K$ is defined in Eq. (21).
[4] Incomplete beta function $\mathcal{B}(x; a, b) = \int_0^x u^{a-1}(1-u)^{b-1}\, \mathrm{d}u$.

with concrete regularizers. Sec. 7 collects practical considerations and limitations of the surrogate-reward viewpoint. Sec. 8 reviews concurrent work and connects our surrogate-reward formulation of RLVR to conditional stochastic optimization in the optimization literature. We conclude in Sec. 9 with an outlook on our perspective and its prospects. The appendix contains deferred derivations and proofs.

**Notation.** We define a training example as a tuple $(x, a)$, where $x$ is the prompt and $a$ is the reference answer. For this example $(x, a)$, let $y_1, \ldots, y_N$ denote $N$ responses generated IID from model $\pi_\theta(\cdot|x)$. Let $r_i := r_{0/1}(y_i, a)$ denote the 0/1 reward of response $i$, and $\widehat{\rho} := \frac{1}{N} \sum_{i \in [N]} r_i$ denote the empirical 0/1 reward. We denote the per-example expected 0/1 reward as $\rho_\theta(x, a) := \mathbb{E}_{y \sim \pi_\theta(\cdot|x)}[r_{0/1}(y, a)]$. When clear from context, we simply write $\rho$ to denote $\rho_\theta(x, a)$. Let $N^+ := \sum_{i \in [N]} r_i$ and $N^- := N - N^+$ denote the number of correct and wrong responses. We use the shorthand $\nabla_i := \nabla_\theta \log \pi_\theta(y_i|x)$ for the log-probability gradient of individual response $y_i$, and for the average log-probability gradient over correct/wrong responses:

$$\widehat{\nabla}_+ := \frac{1}{N^+} \sum_{i:r_i=1} \nabla_\theta \log \pi_\theta(y_i|x) \quad \text{and} \quad \widehat{\nabla}_- := \frac{1}{N^-} \sum_{i:r_i=0} \nabla_\theta \log \pi_\theta(y_i|x) \,. \tag{5}$$

Throughout, $A_i$ denotes the advantage score of the $i$-th response to problem $x$ and we silence its dependence on $x$ when it will be clear from context. Finally, we reserve $K$ to denote the argument of Pass@K and quantities with subscript $K$ denote quantities related to Pass@K reward. In particular, $\rho_{K,\theta}$ denotes the per-example expected Pass@K reward and $\hat{\rho}_K$ its empirical counterpart. Notation specific to different algorithms we study in this paper appear as they are introduced in the text.

## 2 Warm-up: Optimizing the 0/1 Reward

With access to a finite training set $\mathcal{D}$ of problem-answer pairs $(x, a)$, which we call examples, a natural approach to maximize the average (over $\mathcal{D}$) reward is stochastic gradient ascent. At each iteration $t$, we update $\theta_{t+1} = \theta_t + \eta \cdot G(\theta_t; \mathcal{D})$ with learning rate $\eta$ and gradient

$$G(\theta; \mathcal{D}) = \widehat{\mathbb{E}}_{(x,a) \sim \mathcal{D}}[\nabla_\theta \rho_\theta(x, a)] = \widehat{\mathbb{E}}_{(x,a) \sim \mathcal{D}} \mathbb{E}_{y \sim \pi_\theta(\cdot|x)} \left[ r_{0/1}(y, a) \cdot \nabla_\theta \log \pi_\theta(y|x) \right] \,, \tag{6}$$

where $\widehat{\cdot}$ denotes empirical averages and we have applied the log-derivative trick: $\nabla_\theta \pi_\theta(y|x) = \pi_\theta(y|x) \cdot \nabla_\theta \log \pi_\theta(y|x)$. The REINFORCE-style algorithms reviewed below provide different approximations of the per-example expected gradient $G_{0/1}(\theta; (x, a)) = \nabla_\theta \rho_\theta(x, a)$.

**REINFORCE.** Classical REINFORCE (Williams, 1992) approximates the expectation over $y \sim \pi_\theta(\cdot|x)$ in (6) using the empirical average over $N$ sampled responses. This yields the per-example gradient:

$$\widehat{G}_{\text{REINFORCE}}(\theta; (x, a)) = \frac{1}{N} \sum_{i \in [N]} r_i \cdot \nabla_i = \widehat{\rho} \cdot \widehat{\nabla}_+ \,. \tag{7}$$

While simple, this estimator suffers from high variance and intuitively provides poor updates since it entirely ignores wrong responses.

**RLOO.** REINFORCE with Leave-One-Out (RLOO) (Kool et al., 2019; Ahmadian et al., 2024) reduces variance by replacing in Eq. (7) rewards $r_i$ with advantage scores that subtract a leave-one-out baseline:

$$A_i = r_i - \frac{1}{N-1} \sum_{j \neq i} r_j \,.$$

This also introduces learning signal from wrong responses. Since $r_i \in \{0, 1\}$, the advantage scores depend only on whether response $y_i$ is correct or wrong:

$$A^+ = 1 - \frac{N\widehat{\rho} - 1}{N - 1} = \frac{N(1 - \widehat{\rho})}{N - 1} \quad \text{and} \quad A^- = -\frac{N\widehat{\rho}}{N - 1} \,. \tag{8}$$

Ignoring the common factor $N/(N-1)$ across all examples, the RLOO gradient simplifies to:

$$\widehat{G}_{\text{RLOO}}(\theta;(x,a)) = A^+ \cdot \widehat{\rho} \cdot \widehat{\nabla}_+ + A^- \cdot (1-\widehat{\rho}) \cdot \widehat{\nabla}_-$$
$$= \widehat{\rho} \cdot (1-\widehat{\rho}) \cdot \left[\widehat{\nabla}_+ - \widehat{\nabla}_-\right]. \tag{9}$$

This is an unbiased estimator of $\nabla_\theta \rho_\theta(x,a)$ with reduced variance compared to REINFORCE.

**GRPO.** GRPO (Shao et al., 2024) normalizes advantages by their empirical standard deviation:

$$A_i = \frac{r_i - \widehat{\rho}}{\sqrt{\widehat{\rho}(1-\widehat{\rho})}}.$$

This yields correct/wrong advantage scores:[5]

$$A^+ = \frac{1-\widehat{\rho}}{\sqrt{\widehat{\rho}(1-\widehat{\rho})}} = \sqrt{\frac{1-\widehat{\rho}}{\widehat{\rho}}} \qquad \text{and} \qquad A^- = \frac{-\widehat{\rho}}{\sqrt{\widehat{\rho}(1-\widehat{\rho})}} = -\sqrt{\frac{\widehat{\rho}}{1-\widehat{\rho}}}. \tag{10}$$

Thus, in the fully-online (single-update-per-rollout) regime where the PPO-style clipping term is inactive, and ignoring the KL regularization term (see App. A for the full GRPO objective and discussion), the GRPO gradient can be expressed conveniently as (Deng et al., 2025a):

$$\widehat{G}_{\text{GRPO}}(\theta;(x,a)) = \sqrt{\widehat{\rho} \cdot (1-\widehat{\rho})} \cdot \left[\widehat{\nabla}_+ - \widehat{\nabla}_-\right]. \tag{11}$$

Comparing Eqs. (9) and (11), we see both RLOO and GRPO have the form $[\widehat{\nabla}_+ - \widehat{\nabla}_-]$ scaled by a function of $\widehat{\rho}$. RLOO scales by $\widehat{\rho}(1-\widehat{\rho})$, while GRPO uses $\sqrt{\widehat{\rho}(1-\widehat{\rho})}$. Throughout, we call such multiplicative factors the *effective gradient weights* of the respective algorithm.

## 3 Pass@K by Direct Differentiation

In analogy to the 0/1 case, the natural approach to optimize the expected Pass@K reward is stochastic gradient ascent with gradient

$$G_{\text{Pass@K}}(\theta) = \widehat{\mathbb{E}}_{(x,a)\sim\mathcal{D}}[\nabla_\theta \rho_{K,\theta}(x,a)].$$

To compute the gradient, we rewrite the per-example Pass@K reward in terms of the per-example 0/1 reward. Using the IID nature of generated responses:

$$\rho_{K,\theta}(x,a) = \Pr_{\{y_i\}\sim\pi_\theta(\cdot|x)} (\exists i \in [K] : y_i \text{ is correct}) = 1 - \prod_{i\in[K]} \Pr_{y_i\sim\pi_\theta(\cdot|x)} (y_i \text{ is } not \text{ correct}) = 1 - (1-\rho_\theta(x,a))^K.$$

Thus, following Mahdavi et al. (2025): $\nabla_\theta \rho_{K,\theta}(x,a) = K(1-\rho_\theta(x,a))^{K-1} \cdot \nabla_\theta \rho_\theta(x,a) = K(1-\rho_\theta(x,a))^{K-1} \cdot G_{0/1}(\theta;(x,a))$. Now, we recognize that the term in parentheses equals the per-example Fail@(K-1)= $1-$Pass@(K-1) reward. Hence, we obtain the final expression for the per-example (population) gradient:

$$G_{\text{Pass@K}}(\theta;(x,a)) := K \cdot \underbrace{(1-\rho_{K-1,\theta}(x,a))}_{\text{Fail@(K-1)}} \cdot G_{0/1}(\theta;(x,a)). \tag{12}$$

This is a reweighting of the respective gradient $G_{0/1}(\theta;(x,a)) = \mathbb{E}_{y\sim\pi_\theta(\cdot|x)}\left[r_{0/1}(y,a)\nabla_\theta \log \pi_\theta(y|x)\right]$ for the 0/1 reward. For $K=1$, the gradient updates are indeed equivalent.

---

[5]Note that GRPO directly uses mean estimates across all responses (rather than leave-one-out). More importantly, it normalizes by standard deviation, making the gradient a biased estimator of $\nabla_\theta \rho_\theta(x,a)$.

**REINFORCE$_K$.** With access to a finite number of responses per example, we need to approximate $G_{\text{Pass@K}}(\theta; (x,a))$ with an empirical estimate. To arrive at an unbiased estimator, we require an unbiased estimator for the Fail@(K-1) term in Eq. (12). For this, we leverage the following unbiased estimator of $\rho_{K,\theta}(x,a)$ (Chen et al., 2021):

$$\widehat{\rho}_K := \widehat{\rho}_{K,\theta}(x,a) = 1 - \binom{N^-}{K} / \binom{N}{K}, \tag{13}$$

representing the probability of drawing $K$ responses without replacement from $N$ total responses such that at least one is correct. Having unbiased estimators for both terms in Eq. (12), we combine them via the leave-one-out trick to form an unbiased estimator of their product:

$$\widehat{G}_{\text{REINFORCE}_K}(\theta; (x,a)) = \frac{K}{N} \sum_{i \in [N]} (1 - \widehat{\rho}_{K-1}^{\text{loo},i}) \cdot r_i \cdot \nabla_i, \tag{14}$$

where $1 - \widehat{\rho}_{K-1}^{\text{loo},i}$ is the leave-one-out unbiased estimator of Fail@(K-1) excluding the $i$-th response. For binary rewards, simple algebra (deferred to Appendix B.2) shows that

$$1 - \widehat{\rho}_{K-1}^{\text{loo},i} = \begin{cases} \widehat{f}_{K-1}^+ := (1 - \widehat{\rho}_K)/(1 - \widehat{\rho} - \frac{K-1}{N}) & \text{if } y_i \text{ is correct} \\ \widehat{f}_{K-1}^- := (1 - \widehat{\rho}_K)/(1 - \widehat{\rho}) & \text{if } y_i \text{ is wrong} \end{cases}. \tag{15}$$

The weights $\widehat{f}_{K-1}^+$ and $\widehat{f}_{K-1}^-$ are leave-one-out empirical estimates of Fail@(K-1), both expressed in terms of the empirical Pass@K reward $\widehat{\rho}_K$. With these, the REINFORCE$_K$ update takes the following final form:

$$\widehat{G}_{\text{REINFORCE}_K}(\theta; (x,a)) = \widehat{f}_{K-1}^+ \cdot \widehat{\rho} \cdot \widehat{\nabla}_+ = \widehat{f}_{K-1}^+ \cdot \widehat{G}_{\text{REINFORCE}}(\theta; (x,a)). \tag{16}$$

Reweighing the vanilla REINFORCE gradient[6] by $\widehat{f}_{K-1}^+$, the empirical Fail@(K-1) of the other $N-1$ samples, amplifies gradients of examples with rare correct responses (large $N^-$) and suppresses gradients of redundant examples (small $N^-$). When $N^- < K-1$, the weight is 0 since Pass@K is already guaranteed. As in the 0/1 case, REINFORCE is unbiased but high-variance estimator. We can reduce variance by substituting rewards $r_i$ in Eq. (14) with appropriately computed advantages. This way, we also introduce negative gradients.

**RLOO$_K$.** Using the RLOO advantage scores from Eq. (8) in place of $r_i$ in Eq. (14), the RLOO algorithm for Pass@K optimization has gradient (proportional to, ignoring constants $N/(N-1)$ and $K$):

$$\widehat{G}_{\text{RLOO}_K}(\theta; (x,a)) = \widehat{f}_{K-1}^+ \cdot \widehat{\rho}(1 - \widehat{\rho}) \cdot \widehat{\nabla}_+ - \widehat{f}_{K-1}^- \cdot \widehat{\rho}(1 - \widehat{\rho}) \cdot \widehat{\nabla}_-. \tag{17}$$

Note the asymmetric weighting for correct versus wrong responses due to Pass@K maximization. This is an unbiased estimator (see App. B.3 for proof) of $G_{\text{Pass@K}}(\theta; (x,a))$ with lower variance compared to $\widehat{G}_{\text{REINFORCE}_K}(\theta; (x,a))$.

**GRPO$_K$.** Analogously, we obtain a GRPO update for Pass@K optimization by using the GRPO advantage scores from Eq. (10) in place of $r_i$ in Eq. (14):

$$\widehat{G}_{\text{GRPO}_K}(\theta; (x,a)) = \widehat{f}_{K-1}^+ \cdot \sqrt{\widehat{\rho}(1 - \widehat{\rho})} \cdot \widehat{\nabla}_+ - \widehat{f}_{K-1}^- \cdot \sqrt{\widehat{\rho}(1 - \widehat{\rho})} \cdot \widehat{\nabla}_-. \tag{18}$$

## 4 Pass@K by Advantage Shaping

The two Pass@K optimization algorithms, RLOO$_K$ and GRPO$_K$, of Section 3 differ from their vanilla counterparts through *asymmetric* advantage reweighting. Specifically, their effective advantage score is:

$$A_{\text{Pass@K}}^\pm = \widehat{f}_{K-1}^\pm \cdot A^\pm, \tag{19}$$

where $\widehat{f}_{K-1}^\pm$ are defined in Eq. (15), and $A^\pm$ are the vanilla advantage scores (e.g., Eq. (10) for GRPO).

---

[6]We have dropped a constant $K$ factor across all examples, which can be absorbed into the learning rate.

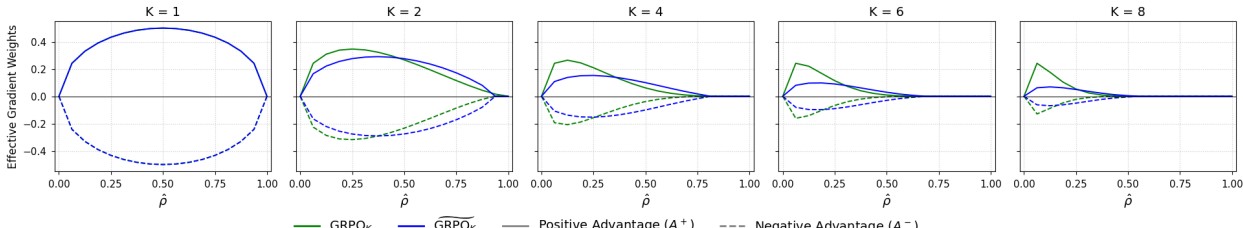

Figure 2: Effective gradient weights of $\text{GRPO}_K$ ($A_K^{\pm}$ in Eq. (19)) vs. $\widetilde{\text{GRPO}}_K$ ($\widetilde{A}_K^{\pm}$ in Eq. (21)). These weights scale the gradients of correct (solid lines) and wrong (dashed lines) responses by a factor $A^+ \cdot \hat{\rho}$ and $A^- \cdot (1 - \hat{\rho})$, respectively. The scores are plotted against the empirical 0/1 rate ($\hat{\rho}$) for a fixed sample size $N = 16$ and varying $K$. For $K = 1$, both weights coincide with vanilla GRPO weights $\sqrt{\hat{\rho}(1 - \hat{\rho})}$ (Eq. (11)). Both methods downweight easy examples (large $\hat{\rho}$), but $\text{GRPO}_K$ is more aggressive and uses asymmetric correct/wrong weights, unlike the symmetric scaling in $\widetilde{\text{GRPO}}_K$.

**$\widetilde{\text{GRPO}}_K$ via advantage shaping.** We now review the GRPO variant proposed by Chen et al. (2025), which we denote $\widetilde{\text{GRPO}}_K$, and which directly shapes GRPO's advantages to favor Pass@K improvements. Their proposed advantage scores, which we denote $\widetilde{A}_{\text{Pass@K}}^{\pm}$, are given in (Chen et al., 2025, App. B3) as:

$$\widetilde{A}_{\text{Pass@K}}^+ = \sqrt{(1 - \widehat{\rho}_K)/\widehat{\rho}_K} \qquad \text{and} \qquad \widetilde{A}_{\text{Pass@K}}^- = (1 - \widehat{\rho}_K - \widehat{f}_{K-1}^-)/\sqrt{\widehat{\rho}_K(1 - \widehat{\rho}_K)}, \tag{20}$$

expressed in our notation of empirical Pass@K reward $\widehat{\rho}_K$ (Eq. (13)) and the leave-one-wrong-out Fail@(K-1) estimator $\widehat{f}_{K-1}^-$ (Eq. (15)). As noted in (Deng et al., 2025b), this can be simplified: From Eq. (15), we have $1 - \widehat{\rho}_K = (1 - \widehat{\rho}) \cdot \widehat{f}_{K-1}^-$. Thus, $\widetilde{A}_{\text{Pass@K}}^- = -\frac{\widehat{\rho}}{1-\widehat{\rho}} \widetilde{A}_{\text{Pass@K}}^+$, and in terms of vanilla GRPO advantage:

$$\widetilde{A}_{\text{Pass@K}}^{\pm} = \widetilde{\omega}_K \cdot A^{\pm}, \qquad \text{where} \quad \widetilde{\omega}_K := \sqrt{\frac{1 - \widehat{\rho}_K}{\widehat{\rho}_K}} \cdot \sqrt{\frac{\widehat{\rho}}{1 - \widehat{\rho}}}. \tag{21}$$

Overall, the per-example gradient of $\widetilde{\text{GRPO}}_K$ is:

$$\widehat{G}_{\widetilde{\text{GRPO}}_K}(\theta; (x,a)) = \widetilde{\omega}_K \cdot \widehat{G}_{\text{GRPO}}(\theta; (x,a)) = \sqrt{\frac{1 - \widehat{\rho}_K}{\widehat{\rho}_K}} \cdot \widehat{\rho} \cdot \left[\widehat{\nabla}_+ - \widehat{\nabla}_-\right]. \tag{22}$$

**Comparing the two approaches.** Both methods reweight vanilla advantages and have a similar effect: compared to the vanilla methods of Sec. 2 that optimize 0/1 reward, they aggressively downweight gradient contributions from examples with medium-to-high empirical 0/1 rewards, thereby favoring more difficult examples. Yet, they differ structurally: The reweighting in Eq. (21) is *symmetric* (same multiplier $\widetilde{\omega}_K$ for correct and wrong responses), whereas Eq. (19) uses *asymmetric* weights ($\widehat{f}_{K-1}^+ \neq \widehat{f}_{K-1}^-$). Moreover, direct optimization more aggressively dampens contributions when the empirical reward is high (and amplifies more when low). Figure 2 visualizes the effective weights each algorithm applies to correct vs. wrong gradients.

## 5 Bridging the Views

### 5.1 Reverse-Engineering: Surrogate Reward from Advantage Shaping

Section 4 showed that both Pass@K methods, direct optimization ($\text{GRPO}_K$) and advantage shaping ($\widetilde{\text{GRPO}}_K$), can be expressed as reweightings of vanilla GRPO. We now tighten this connection by showing that advantage shaping also admits an interpretation as direct optimization of surrogate Pass@K reward.

**Claim 1** (Reverse-Engineering $\widetilde{\text{GRPO}}_K$)**.** *The $\widetilde{\text{GRPO}}_K$ algorithm by Chen et al. (2025) rescales the vanilla GRPO empirical gradient, yielding updates as shown in Eq. (22). For sufficiently large sample size $N \gg K$,*

*this corresponds to direct maximization of the per-example surrogate reward*

$$\frac{2}{K} \arcsin\left( \sqrt{\rho_{K,\theta}(x,a)} \right),$$ (23)

*where $\rho_{K,\theta}(x,a) := \mathbb{E}_{\{y_i\}_{i \in [K]} \sim \pi_\theta(\cdot|x)} \, r_{Pass@K}(\{y_i\}_{i \in [K]}, a)$ is the expected Pass@K reward per example.*

Thus, the advantage-shaping approach of Chen et al. (2025) implicitly optimizes a smooth, differentiable transformation of the Pass@K reward, providing further justification for the validity of the different approach by which the authors arrive at it. Note that the surrogate reward is maximized when the Pass@K reward equals 1, and is actually strictly monotone, so both the original and surrogate objectives share the same optimal solution, though the arcsin surrogate reward could lead to a different optimization path.

**Connection to variance stabilizing transforms.** The arcsin transform that emerges is noteworthy: it is a *variance-stabilizing transformation* (VST) for the binomial distribution. Specifically, for $\mathcal{X} \sim \mathrm{Bin}(M, p)$, the arcsin transform satisfies $\mathrm{Var}(\sqrt{M} \cdot \arcsin(\sqrt{\mathcal{X}/M})) \approx 1/4$ independent of $p$ (Anscombe, 1948). This connection is not coincidental: Chen et al. (2025, Sec. 2.2) arrived at their method through arguments using batched sampling, where $N$ responses are partitioned into $N/K$ groups of size $K$. In this setting, the number of successful groups follows a binomial distribution with $M = N/K$ trials, and the arcsin transform would stabilize the variance of this estimate. Formalizing a potential the connection between VSTs and optimization stability when applying empirical gradients to transformed rewards is open direction for future work.

## 5.2 Forward-Engineering: Advantage Shaping from Surrogate Reward

$\widetilde{\mathrm{GRPO}}_K$ can be reverse-engineered as optimizing an implicit population-level surrogate reward. Now, we address the forward direction: *How to arrive at $\widetilde{\mathrm{GRPO}}_K$ starting from the arcsin-transform surrogate reward?*

**Claim 2** ($\widetilde{\mathrm{GRPO}}_K$=RLOO on Surrogate Reward). *The $\widetilde{\mathrm{GRPO}}_K$ policy-gradient update by Chen et al. (2025) is equivalent to an RLOO-style policy gradient update to the surrogate per-example reward $\frac{2}{K} \arcsin\left( \sqrt{\rho_{K,\theta}(x,a)} \right)$, from Eq. (23).*

To see this, we work as in Sec. 3, but now with the arcsin-transformed objective. We proceed in two steps: (i) By direct differentiation, the population per-example gradient is

$$\frac{2}{K} \nabla_\theta \arcsin\left( \sqrt{\rho_{K,\theta}} \right) = \frac{1}{K} \frac{1}{\sqrt{\rho_{K,\theta}(1-\rho_{K,\theta})}} \nabla_\theta \rho_{K,\theta} = \frac{1}{\sqrt{\rho_{K,\theta}(1-\rho_{K,\theta})}} (1-\rho_\theta)^{K-1} \nabla_\theta \rho_\theta$$

$$= \frac{1}{\sqrt{\rho_{K,\theta}(1-\rho_{K,\theta})}} \cdot \frac{1-\rho_{K,\theta}}{1-\rho_\theta} \cdot \nabla_\theta \rho_\theta,$$ (24)

(ii) To obtain an empirical version, we (a) approximate the multiplicative term of the gradient by substituting $\rho_{K,\theta}$ and $\rho_\theta$ with their empirical counterparts $\widehat{\rho}_K$ and $\widehat{\rho}$, and (b) approximate the expected 0/1 reward gradient term with the RLOO gradient update from Eq. (9) (recall Eq. (2)). The update then becomes:

$$\frac{1}{\sqrt{\widehat{\rho}_K(1-\widehat{\rho}_K)}} \cdot \frac{1-\widehat{\rho}_K}{1-\widehat{\rho}} \cdot \widehat{\rho}(1-\widehat{\rho}) \cdot \left[ \widehat{\nabla}_+ - \widehat{\nabla}_- \right] = \sqrt{\frac{1-\widehat{\rho}_K}{\widehat{\rho}_K}} \cdot \widehat{\rho} \cdot \left[ \widehat{\nabla}_+ - \widehat{\nabla}_- \right],$$

and we recognize $\widehat{G}_{\widetilde{\mathrm{GRPO}}_K}(\theta; (x,a))$ in the final expression (see Eq. (22)).

**Alternative surrogate rewards.** This connection invites exploration of alternative variance-stabilizing transformations. The arcsin transform is a VST for the binomial distribution, which aligns with the batched sampling scheme of Chen et al. (2025, Sec. 2.2). However, the standard combinatorial Pass@K estimator does not strictly follow a binomial distribution. Also, the analysis above assumes $N$ grows large while $K$ remains constant. Designing surrogate rewards that account for joint scaling of $N$ and $K$ is an interesting direction, particularly for the practically relevant small-$N$ regime where computational budgets are limited.

**Biased versus unbiased estimation.** In the derivation of Claim 2, contrary to our construction of $\text{RLOO}_K$ in Sec. 3, we did not insist on maintaining unbiasedness of the gradient estimator. $\frac{1-\widehat{\rho}_K}{1-\widehat{\rho}}$ is a biased estimator of the multiplicative term in Eq. (24), and even replacing $\widehat{\rho}_K$ with a leave-one-out estimate would not remove this bias since the product of two unbiased estimators is generally biased. However, note that even the strong vanilla GRPO baseline does not implement an unbiased empirical estimate of the gradient.

**Special case: GRPO.** In fact, applying the logic of Claim 2 to the $K = 1$ case provides an alternative interpretation to GRPO's normalization by the reward's standard deviation.

**Corollary 1** (GRPO as Surrogate Reward Optimization)**.** *The GRPO update (Eq.* (11)*) by Shao et al.* *(2024) is equivalent to an RLOO-style policy gradient update for the surrogate per-example reward*

$$2\arcsin\left(\sqrt{\rho_\theta(x,a)}\right)$$

*where $\rho_\theta(x,a) := \mathbb{E}_{y\sim\pi_\theta(\cdot|x)}\, r_{0/1}(y,a)$ is the per-example expected 0/1 reward.*

Indeed, the population gradient of this surrogate is $\nabla_\theta\rho/\sqrt{\rho(1-\rho)}$. Approximating the leading multiplicative factor by $1/\sqrt{\widehat{\rho}(1-\widehat{\rho})}$ and applying an RLOO-style estimator for $\nabla_\theta\rho$ (which is proportional to $\widehat{\rho}(1-\widehat{\rho})[\widehat{\nabla}_+ - \widehat{\nabla}_-]$ by Eq. (9)) yields update $\sqrt{\widehat{\rho}(1-\widehat{\rho})}[\widehat{\nabla}_+ - \widehat{\nabla}_-]$, matching that of vanilla GRPO.

This surrogate reward perspective should be distinguished from GRPO's original formulation. As presented by Shao et al. (2024), GRPO maximizes a PPO-style objective designed for the multi-epoch off-policy setting, where the expectation is over the old policy. Our surrogate reward, in contrast, identifies the on-policy population objective that the GRPO gradient ascends in the fully online setting. We detail this distinction in App. A, showing why the PPO-style objective cannot directly yield this on-policy surrogate.

## 5.3 Equivalence of Advantage Shaping and Surrogate Reward

The surrogate-reward perspective developed in the previous part of this section for $\widetilde{\text{GRPO}}_K$ (and its special case vanilla GRPO for $K = 1$) applies more generally to a broad class of binary-reward RLVR policy gradient algorithms whose per-example gradient can be expressed as:

$$\widehat{G}(\theta;(x,a)) = \omega_+(\widehat{\rho})\cdot\widehat{\nabla}_+ - \omega_-(\widehat{\rho})\cdot\widehat{\nabla}_- \,, \tag{25}$$

for some functions $\omega_\pm : [0,1] \to \mathbb{R}$ which determine the effective gradient weights multiplying the average empirical gradients of correct and incorrect responses respectively. This form corresponds to the middle column (Weighted Empirical Gradient) of Table 1; as seen, all algorithms we study here fit this expression.

The surrogate reward perspective provides an interpretation of any such algorithm in the population limit of large $N$. Specifically, the following claim generalizes Claim 1.

**Claim 3.** *In the population limit $N \to \infty$, a policy-gradient algorithm with per-example gradient of the form* (25) *performs gradient ascent on a per-example surrogate reward $F(\rho_\theta(x,a))$, where $F : [0,1] \to \mathbb{R}$ is differentiable and satisfies for $u \in (0,1)$:*

$$F'(u) \;=\; \frac{\omega_+(u)}{u} + \frac{\omega_-(u)}{1-u}.$$

*Consequently, the update averaged over examples targets maximizing the population surrogate objective*

$$\mathbb{E}_{(x,a)\sim\mathcal{P}}\Big[\, F\big(\mathbb{E}_{y\sim\pi_\theta(\cdot|x)}r(y,a)\big)\,\Big]. \tag{26}$$

*This objective is a surrogate transformation of the conventional RLVR objective in Eq.* (1)*.*

*Proof.* Replacing empirical quantities $\widehat{\cdot}$ with their population counterparts, the population-level per-example gradient is

$$G(\theta;(x,a)) \;=\; \omega_+(\rho_\theta)\cdot\mathbb{E}_{y\sim\pi_\theta(\cdot|x)}[\nabla_\theta\log\pi_\theta(y|x)\,|\,r(y,a)=1] - \omega_-(\rho_\theta)\cdot\mathbb{E}_{y\sim\pi_\theta(\cdot|x)}[\nabla_\theta\log\pi_\theta(y|x)\,|\,r(y,a)=0]\,,$$

where we use our shorthand $\rho_\theta = \rho_\theta(x, a)$. Evaluating the conditional expectations (Lemma 2 in App. B.1) yields

$$G(\theta; (x, a)) = \omega_+(\rho_\theta) \cdot \frac{\nabla_\theta \rho_\theta}{\rho_\theta} + \omega_-(\rho_\theta) \cdot \frac{\nabla_\theta \rho_\theta}{1 - \rho_\theta} = \left[ \frac{\omega_+(\rho_\theta)}{\rho_\theta} + \frac{\omega_-(\rho_\theta)}{1 - \rho_\theta} \right] \nabla_\theta \rho_\theta =: \ F'(\rho_\theta) \cdot \nabla_\theta \rho_\theta.$$

Thus, in the population limit, the algorithm updates are of the form

$$\theta_{t+1} \ = \ \theta_t + \eta \cdot F'(\rho_\theta) \cdot \nabla_\theta \rho_\theta, \tag{27}$$

which is gradient ascent on the surrogate reward $F(\rho_\theta)$ (up to an additive constant in $F$). □

**Examples (reverse-engineering).**  For $\omega_-(u) = \omega_+(u) = u(1-u)$ (RLOO, Eq. (9)), we obtain $F'(u) = 1$, hence $F(u) = u$ (up to a constant). Similarly, for $\omega_+(u) = u$ and $\omega_-(u) = 0$ (REINFORCE, Eq. (7)), we also have $F'(u) = 1$ and thus $F(u) = u$. For GRPO, where $\omega_-(u) = \omega_+(u) = \sqrt{u(1-u)}$ (Eq. (11)), we have $F'(u) = \frac{1}{\sqrt{u(1-u)}}$ and thus $F(u) = 2 \arcsin(\sqrt{u})$ (up to a constant), matching Corollary 1.

Conversely, starting from a differentiable surrogate reward $F : [0, 1] \to \mathbb{R}$, we can derive an RLVR policy-gradient update that (in the population limit) performs gradient ascent on $F(\rho_\theta)$. At the population level, gradient ascent corresponds exactly to (27). To obtain a practical finite-$N$ algorithm, [7] we replace population quantities by empirical analogues: (a) replace $F'(\rho_\theta)$ with $F'(\widehat{\rho})$, where $\widehat{\rho}$ is the empirical (finite-$N$) estimate of $\rho_\theta$; and (b) replace $\nabla_\theta \rho_\theta$ with its RLOO proxy $\widehat{G}_{\text{RLOO}}(\theta) = \widehat{\rho}(1 - \widehat{\rho}) \cdot (\widehat{\nabla}_+ - \widehat{\nabla}_-)$ (Eq. (2)).

This yields the surrogate-driven finite-$N$ gradient estimator

$$\widehat{G}_F(\theta; (x, a)) \ = \ \omega(\widehat{\rho}) \cdot (\widehat{\nabla}_+ - \widehat{\nabla}_-), \qquad \text{where} \quad \omega(u) := F'(u) \cdot u \cdot (1 - u).$$

Equivalently, in the conventional advantage form,

$$\widehat{G}_F(\theta; (x, a)) = \frac{1}{N} \sum_{i \in [N]} A_i^F \cdot \nabla_i, \qquad A_i^F := F'(\widehat{\rho}) \cdot (r_i - \widehat{\rho}) = \begin{cases} F'(\widehat{\rho}) \cdot (1 - \widehat{\rho}) & r_i = 1, \\ -F'(\widehat{\rho}) \cdot \widehat{\rho} & r_i = 0. \end{cases} \tag{28}$$

Up to clipping and KL regularization, (28) is a GRPO-style update in which the surrogate choice $F$ shapes (i.e., reweighs) the binary learning signal through the scalar factor $F'(\widehat{\rho})$. It can be easily checked that

$$A_i^F = F'(\widehat{\rho}) \sqrt{\widehat{\rho}(1 - \widehat{\rho})} \cdot A_i^{\text{GRPO}},$$

with $A_i^{\text{GRPO}}$ the vanilla GRPO advantage scores in Eq. (10).

**Examples (forward-engineering).**  Letting $F(u) = u$ recovers (up to a constant scaling $N/(N-1)$) the vanilla RLOO update. Letting $F(u) = 2 \arcsin(\sqrt{u})$ recovers the GRPO advantage shaping (Corollary 1). For Pass@K optimization, letting $F(u) = 1 - (1-u)^K$ yields $\text{RLOO}_K$, while letting $F(u) = \frac{2}{K} \arcsin\left(\sqrt{1 - (1-u)^K}\right)$ yields $\widetilde{\text{GRPO}}_K$ (Claim 2). As another example, letting $F(u) = q^{-1} u^q$ for any $q > 0$ yields a one-parameter family of strictly increasing surrogate objectives with $F'(\widehat{\rho}) = \widehat{\rho}^{q-1}$, i.e., $\omega(\widehat{\rho}) = \widehat{\rho}^q (1 - \widehat{\rho})$: $q < 1$ upweights low-$\widehat{\rho}$ examples, while $q > 1$ upweights high-$\widehat{\rho}$ examples.

In all these cases $F$ is strictly increasing: $F(\rho_\theta)$ is maximized at 1 just like $\rho_\theta$, but induces possibly different optimization dynamics via shaped advantages. In the following section, we provide additional examples of surrogate objectives that yield new RLVR algorithmic instances while possibly relaxing the requirement of the optimization objective being strictly increasing. We then conclude with practical remarks in Sec. 7, including the generally biased nature of the finite-$N$ update (28), and how one can incorporate clipping and KL regularization to obtain practical GRPO-like algorithms in offline settings (at the cost of sacrificing the exact surrogate-maximization guarantee). We also invite the community to explore alternative choices of $F$ and their potential practical benefits.

---

[7]This empirical substitution represents one practical algorithmic choice. While highly sensible and consistent within our analysis (for instance, when applied to $F(u) = 2 \arcsin(\sqrt{u})$ it exactly recovers the strong GRPO baseline), it is not the only option. Alternative gradient estimators, such as those leveraging variance reduction or debiasing techniques from the conditional stochastic optimization literature (see Sec. 8), could also be employed to optimize the same underlying surrogate reward.

# 6  A Regularized Surrogate Rewards Perspective

Both Pass@K routes examined in Secs. 3 and 4 tilt the learning signal toward the hard cases. Chen et al. (2025) argue this is aligned with the exploit/explore roles of the two metrics. Maximizing the 0/1 reward emphasizes *exploitation*: making individual responses more often correct, even on already-easy examples. In contrast, Pass@K encourages *exploration*: for a given example, only one of $K$ responses must be correct, so once an example is "solved" in the Pass@K sense, further pushing its 0/1 reward is less valuable than allocating gradient signal to unsolved examples where more search could pay off. Ideally, we want a controlled balance: retain sufficient signal on solved examples while injecting additional search signal on unsolved ones. To achieve this, they propose mixing Pass@K-shaped and vanilla advantages.

Using reverse-engineering, we arrive at a complementary interpretation of their method as *regularized surrogate reward maximization.* This perspective decomposes the RL objective into a data-fitting term (the transformation of the mean reward that the GRPO baseline optimizes) and a regularizer (an uncertainty term such as the reward's standard deviation). While the data-fitting term can be interpreted as exploitation and the regularizer as exploration, the regularization viewpoint itself provides a possibly more elementary justification for practical advantage-shaping heuristic-modifications of GRPO.

## 6.1  Combined 0/1 and Pass@K Advantage Shaping

Having observed that Pass@K optimization can *aggressively* favor difficult examples (small $\widehat{\rho}$) while potentially zeroing out useful signal on easier ones (medium/large $\widehat{\rho}$), Chen et al. (2025) propose a simple blend between optimizing the 0/1 reward and optimizing Pass@K. They combine the Pass@K advantage with the vanilla advantage using the empirical 0/1 reward $\widehat{\rho}$ as a mixing weight. In our notation:

$$\widetilde{A}^{\pm}_{\text{mix}} \;=\; \widehat{\rho}\,\widetilde{A}^{\pm}_{\text{Pass@K}} \;+\; (1-\widehat{\rho})\,A^{\pm} \;=\; \left(1 - \widehat{\rho} + \widehat{\rho}\cdot\widetilde{\omega}_K\right) A^{\pm}, \tag{29}$$

where we used Eq. (21) to write $\widetilde{A}^{\pm}_{\text{Pass@K}} = \widetilde{\omega}_K\,A^{\pm}$. The same idea can be instantiated for the *direct* Pass@K optimization view presented in Sec. 3. Since $A^{\pm}_{\text{Pass@K}} = \widehat{f}^{\pm}_{K-1}\,A^{\pm}$ by (19), an analogous combination yields

$$A^{\pm}_{\text{mix}} \;=\; \widehat{\rho}\,A^{\pm}_{\text{Pass@K}} \;+\; (1-\widehat{\rho})\,A^{\pm} \;=\; \left(1 - \widehat{\rho} + \widehat{\rho}\cdot\widehat{f}^{\pm}_{K-1}\right) A^{\pm}. \tag{30}$$

Figure 3 visualizes the resulting *effective* gradient weights (i.e., $\widehat{\rho}A^{+}_{\text{mix}}$ for correct and $(1-\widehat{\rho})|A^{-}_{\text{mix}}|$ for wrong responses). Observe this blending avoids the overdamping of mid/large-$\widehat{\rho}$ training signal that can occur under pure Pass@K optimization. Interestingly, the two combinations become numerically very close as $K$ grows. Moreover, except when $K$ is small relative to $N$, both essentially collapse to the lower envelope $(1 - \widehat{\rho})A^{\pm}$. That is, they essentially no longer exploit an explicit Pass@K signal.

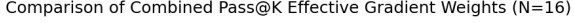

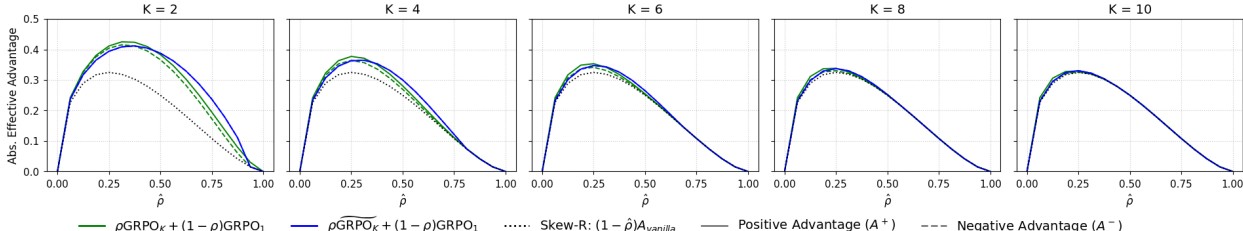

Figure 3: Combined Pass@K *effective* gradient scores for $N = 16$ and various $K$. Green uses Eq. (30); Blue uses Eq. (29); Dotted gray is the skew-R $(1-\widehat{\rho})A^{\pm}$. For $K = 2,4$ the two combinations differ slightly at mid/high $\widehat{\rho}$ (blue is modestly larger); for $K \geq 6$ they nearly coincide across $\widehat{\rho}$. This is because, for most $\widehat{\rho}$, the Pass@K scalers vanish for $\widehat{\rho} < 1 - K/N$, reducing both combinations to the same envelope $(1-\widehat{\rho})\sqrt{\widehat{\rho}(1-\widehat{\rho})}$.

## 6.2  Reward Regularization Interpretation of Skew-R

*What reward does the combined 0/1 and Pass@K advantage shaping actually optimize?* Because Sec. 4 connects advantage shaping to direct Pass@K optimization, one might guess it maximizes a convex combination

of the 0/1 and Pass@K rewards. The catch is that in (29) the mixing weight $\widehat{\rho}$ is *adaptive* to the instance "difficulty" (as measured by $\widehat{\rho}$), so the combination is *not* a fixed convex blend at the objective level.

To simplify reverse engineering and obtain a more interpretable surrogate reward, we leverage the empirical observation from the previous subsection: except when $K$ is small relative to $N$, both combined schemes $\widetilde{A}_{\text{mix}}^{\pm}$ and $A_{\text{mix}}^{\pm}$ are numerically close in terms of effective weights to the simpler *right-skew* shaping

$$A_{\text{skew-R}}^{\pm} := (1 - \widehat{\rho})\, A^{\pm}.$$

**Claim 4** (Skew-R= Regularized Surrogate Reward Maximization). *The skew-R (pronounced "skewer") method rescales the vanilla 0/1 advantage by $(1 - \widehat{\rho})$, yielding gradient updates $\widehat{G}_{skew\text{-}R}(\theta; (x, a)) = (1 - \widehat{\rho}) \cdot \widehat{G}_{GRPO}(\theta; (x, a))$. When $N \gg 1$, this corresponds to maximization of the per-example reward*

$$\arcsin(\sqrt{\rho}) + \sqrt{\rho(1 - \rho)}, \tag{31}$$

*where $\rho := \rho_\theta(x, a) := \mathbb{E}_{y \sim \pi_\theta(\cdot|x)}[r_{0/1}(y, a)]$ is the (population) 0/1 reward per example.*

**Reward-regularization perspective.** Vanilla REINFORCE and RLOO maximize the expected 0/1 reward $\rho$ (pure exploitation). As we saw in Corollary 1, vanilla GRPO maximizes a strictly monotone transform of the mean, $\arcsin(\sqrt{\rho})$, so it has the *same maximizers* as $\rho$ and likewise corresponds to exploitation. In contrast, the skew-R reward can be interpreted as stochastic gradient ascent on a regularized surrogate reward that combines a data-fitting term and a regularizer. The data-fitting term incentivizes finding a policy that maximizes the reward over the training examples $(x, a) \in \mathcal{D}$. In contrast, the regularizer alone is maximized at $\rho = 1/2$ where the binary correct and wrong policy outcomes are equally probable. An intuitive explanation of this term, aligning with the regularization effect in classical supervised learning, is that it incentivizes the model to not "overfit" on the training examples, maintaining diversity in wrong response that may explore different solution paths generalizing to unseen examples. Formalizing such a notion of overfitting and its connection to regularization could provide deeper insights into RLVR learning tradeoffs. We note that the argmax of the combined reward in Eq. (31) is still at $\rho = 1$, reassuring here that we incentivize ultimately finding parameters that maximize the average reward. However, mirroring lessons from supervised fine-tuning, this might not always be the best strategy for generalization, and stronger regularization with an explicit parameter controlling the strength could be beneficial (see next subsection).

**Implications for the "Prioritized Sampling" strategy of Kimi 1.5.** An interesting observation potentially supporting the practical value of skew-R: Kimi 1.5 (Team et al., 2025) performs priority sampling in their RL experiments, reweighting each example by $1 - \widehat{\rho}$ to allow harder examples to appear more frequently. This corresponds to the advantage shaping of skew-R (Eq. (31)) applied at the example level rather than within gradient computation. Thus, our interpretation as regularized reward maximization provides a direct justification for their empirically motivated practical choice.

**Consistency of interpretations.** In the large-sample limit $N \gg 1$, skew-R is equivalent to $\text{GRPO}_{K=2}$. In Appendix C.2, we elaborate on this comparison with emphasis on interpreting the reverse-engineered surrogate reward. We show that $\text{GRPO}_{K=2}$ effectively optimizes an incomplete-beta-function transformation of the Pass@2 ($\rho_2$) reward, $\mathcal{B}(1 - (1 - \rho_2)^{1/2}; 1/2, 3/2)$, and that this objective is algebraically identical to the skew-R surrogate in Eq. (31). Thus, in the population limit, both methods are in fact maximizing the same scalar objective. This validates our surrogate-reward reverse-engineering framework, but it also highlights a subtle interpretive question about what constitutes "regularization." The interpretation depends on which performance metric we take as the target: (1) *Pass@K target.* If Pass@K ($\rho_K$) is treated as the true performance metric, then $\text{GRPO}_K$ should be viewed as directly maximizing a smooth, monotone surrogate of $\rho_K$. In this view there is no extra penalty term: we are simply doing surrogate maximization of the evaluation metric itself. (2) 0/1 *target.* In contrast, if 0/1 reward is the ultimate objective (as with skew-R), the interpretation changes. The baseline GRPO already optimizes a surrogate $\arcsin(\sqrt{\rho})$, and the extra multiplicative $(1 - \widehat{\rho})$ reweighting is interpreted as adding a *regularizer term* to that baseline.

## 6.3 An Example of Reward-Level Regularization

To further illustrate the use of reward regularization, we show how to optimize a surrogate reward that combines the data-fitting term with an entropy regularizer $H(\rho) = -\rho \log(\rho) - (1 - \rho) \log(1 - \rho)$ of the binary

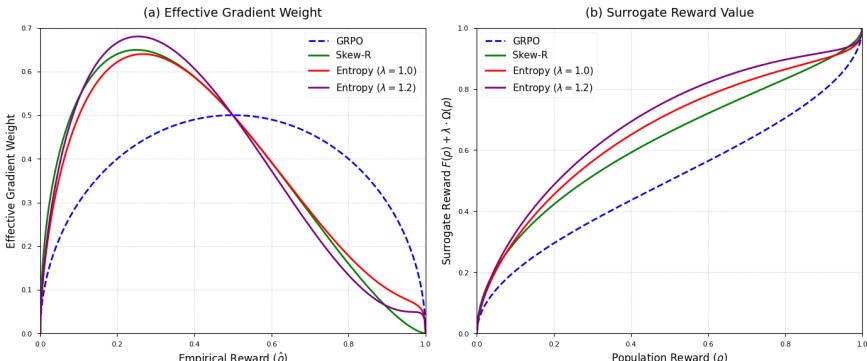

Figure 4: Comparison of (regularized) surrogate objectives (GRPO, Skew-R, and Entropy-Augmented). *(a) Effective Gradient Weight:* Empirical gradient's scaling factor as a function of the empirical reward $\hat{\rho}$. The vanilla GRPO (blue, dashed) is symmetric. Skew-R (green) and the Entropy-augmented methods (red, purple) are asymmetric, suppressing gradients for high $\hat{\rho}$. Skew-R gradients are multiplied here by 2, so all four curves have the same area under the curve. *(b) Surrogate Reward Value:* Corresponding surrogate reward functions, normalized so that they evaluate to 1 at $\rho = 1$. Takeaway: these regularized surrogates prioritize improving low/mid-reward prompts. The regularization strength (larger $\lambda$) controls how strongly updates are shifted away from high-$\hat{\rho}$ prompts, while the regularizer type controls the *shape* of this reweighting (e.g., entropy retains a small but nonzero weight even at high $\hat{\rho}$).

0/1 reward. Intuitively, similar to the standard-deviation regularization of skew-R, $H(\rho)$ favors examples where the model is still "undecided," redistributing gradient effort toward those rather than repeatedly polishing ones it already gets right. The overall surrogate reward is $2\arcsin(\sqrt{\rho}) + \lambda \cdot H(\rho)$. Applying the forward-engineering recipe of Sec. 5.3 yields a tunable advantage-shaping rule that scales the vanilla GRPO gradient as follows:

$$\widehat{G}_{\text{entropy}} = \left[1 - \lambda \cdot \sqrt{\widehat{\rho}(1 - \widehat{\rho})}\log\left(\frac{\widehat{\rho}}{1 - \widehat{\rho}}\right)\right] \cdot \widehat{G}_{\text{GRPO}}. \tag{32}$$

This mirrors the structure of skew-R, but with a different weighting function. Whereas skew-R uses $(1 - \hat{\rho})$ and therefore drives the weight for already-solved examples ($\hat{\rho} \approx 1$) essentially to zero, the entropy-based weight retains a significantly reduced, but nontrivial, weight even for easy prompts, while still sharply boosting harder examples. See Fig. 4 for illustration. Note also that the entropy-augmented objective *need not* be globally maximized at $\rho = 1$ for all $\lambda$: For $\lambda \lesssim 1.5$, its maximizer remains at $\rho = 1$, so we are still ultimately pushing toward perfect success. For larger regularization strengths, the entropy term is strong enough that the overall maximizer shifts to some $\rho < 1$, providing a mechanism to penalize perfect fitting of the training data, a stronger form of regularization.

## 7 Practical Considerations

**Biased vs Unbiased Scalings.** In Sec. 3 we derived Pass@K advantages by using a leave-one-out construction to obtain unbiased estimates of the two components in (12): Fail@$(K-1)$ and $G_{0/1}(\theta; (x, a))$, resulting in the unbiased empirical gradient $\widehat{G}_{\text{RLOO}_K}$ in Eq. (17). In contrast, Mahdavi et al. (2025) apply a *biased* empirical gradient with advantage scaling $(1 - \widehat{\rho})^{K-1} \cdot A^{\pm}$. This remains biased even if $A^{\pm}$ correspond to an unbiased method, such as RLOO. Reverse-engineering their biased estimator shows that it maximizes the same surrogate reward as our unbiased $\text{RLOO}_K$. In fact, the advantage scores resulting from our forward-engineering of surrogate rewards are all biased. Thus, while the surrogate reward provides a unifying perspective to advantage scores, it does not differentiate between biased vs unbiased implementations. While unbiasedness is intuitively a desirable property, see App. C.3 for a discussion on regimes where biased variants can be attractive. Finally, we remark that when the number of responses $N$ per prompt is small, plug-in estimates such as $\hat{\rho}$ (or $\hat{\rho}_K$) can be noisy, and the resulting reweightings may deviate from

their large-sample forms assumed by our surrogate-reward interpretations. Characterizing these finite-$N$ effects is an important direction for future work; we view the surrogate-reward lens as still useful here as an organizing principle that identifies the population objective being approximated.

**Normalization Factors.** The main difference of GRPO from classical RLOO is, up to the clipping, the normalization with respect to the reward's standard deviation. Cor. 1 provides an alternative interpretation of this scaling: at the level of the training objective, it maximizes a transformation of the 0/1 reward that stabilizes the variance of its empirical estimator. More broadly, our forward-engineering of reward surrogates $F(\rho)$ uses RLOO empirical gradients as a robust estimate of the population reward gradient $\nabla_\theta \rho$ and scales them by $F'(\rho)$ that generalizes the specific GRPO normalization via standard deviation. On the empirical side, there is ongoing debate about whether such normalizations are necessary for good performance. For example, in support of earlier works (Ahmadian et al., 2024), a recent study by Khatri et al. (2025) finds that RLOO is competitive to GRPO, with zero-mean advantages being the main driver behind its success (Xiong et al., 2025). This raises an interesting question: Could we theoretically characterize how properties of a surrogate $F$ relate to favorable optimization dynamics? Such an analysis might shed light on if, when and why these normalizations matter in practice.

**Clipping.** Our analysis omits the clip operation of GRPO (Shao et al., 2024), which is itself inherited from PPO (Schulman et al., 2017). This is accurate when GRPO operates in full online mode (single update per model generation), as originally done by Shao et al. (2024). However, if multiple updates are performed per generation, the GRPO gradient expression in Eq. (11) that our analysis uses is only a proxy. That said, current systems are often optimized to be on-policy (Liu et al.; Qi et al., 2025). Either way, all advantage-shaping methods studied here (e.g., $\mathrm{GRPO}_K$, skew-R, entropy-augmented) can be empirically applied in practical, multi-epoch training regimes by simply substituting the respective advantage scores into the original GRPO formulation with the clip operation and/or KL regularization (or follow-up variants, e.g., (Yu et al., 2025a; Zheng et al., 2025)). Specifically, using a surrogate $F$ is essentially a *one-line change* in existing GRPO code: following (28), set $A_i \leftarrow F'(\hat\rho)\sqrt{\hat\rho(1-\hat\rho)}\,A_i^{\mathrm{GRPO}}$ (with $\hat\rho$ the batch success rate) before clipping, then apply the usual clipping/KL.

**Granular Advantage Shaping.** The methods we study shape the advantage at the example level. This is the case for all methods that result from forward-engineering a (regularized) reward surrogate (as in Eq. (4)): they scale vanilla RLOO by a factor $F'(\hat\rho) + \lambda \cdot \Omega'(\hat\rho)$ that depends only on the per-example empirical 0/1 reward $\hat\rho$. The only exception is the Pass@K methods designed in Sec. 3, which introduce asymmetric weights for correct versus wrong responses. Here, advantage shaping is done at the response level. [8] Finer-grained shaping is also possible. For example, Deng et al. (2025b); Cheng et al. (2025); Cui et al. (2025) investigate token-level advantage shaping and would be interesting to reverse-engineer those as regularized surrogate rewards, albeit in the form of finer-grained regularization at the policy/model-representation level.

**Empirical evaluation and surrogate selection.** Beyond providing a unifying lens for interpreting existing RLVR/GRPO-style policy-gradient methods, the established bi-directional relation "advantage shapes $\leftrightarrow$ weighted gradients $\leftrightarrow$ surrogate reward" offers an algorithmic opportunity: a *forward-engineering recipe* for designing advantage-shaping rules. As noted above, the updates derived from this recipe are *plug-and-play* in existing GRPO implementations. In Secs. 5–6, we introduce concrete surrogates as starting points, and in App. D we provide proof-of-concept experiments comparing the performance of variants like Skew-R, $\mathrm{GRPO}_K$, and entropy-regularized methods against vanilla GRPO. For designing future algorithms, the most convenient objects to work with are the surrogate reward $F(\rho_\theta)$ or the associated weighting functions $\omega_\pm(\rho_\theta)$, as these directly specify how the update reweights correct vs. incorrect responses (see Sec. 5.3). Conceptually, the equivalence "weighted gradients $\leftrightarrow$ surrogate reward" mirrors the duality in supervised learning between modifying gradients and modifying loss functions. This suggests that well-studied robust or imbalance-aware loss-design principles from supervised learning can serve as starting points for designing corresponding RLVR updates. Overall, the recipe is intentionally general; a systematic empirical study of this design space that involves identifying robust families across tasks/compute regimes and characterizing trade-offs is an important next step, which we leave to future work. Combined with the theoretical unification through surrogate

---

[8]In this specific case, the weights of correct ($\widehat{f}_{K-1}^+$) and wrong gradients ($\widehat{f}_{K-1}^-$) have the same population limit. The reverse-engineering approach of Claims 1, 4 can still be applied even if these limits are themselves asymmetric (see Appendix C.4).

rewards, such a study may also yield concrete guidelines for when properties of the data distribution, model capacity, or task structure favor one algorithm over another.

## 8 Related Work

Our work is motivated by the rapid development over the past year of variants of the GRPO algorithm, from vanilla binary reward maximization to Pass@K optimization. These developments typically result from selecting appropriate advantage scores, as detailed in the preceding sections. Our work provides a complementary perspective via weighted gradients and surrogate rewards, unifying existing algorithms and offering an alternative recipe for engineering new ones that is more akin to loss design and gradient reweighting in supervised learning (see discussion in Sec. 9).

Contemporaneous work by Davis & Recht (2025) independently arrives at the same reverse-engineering of algorithms such as GRPO as surrogate-reward maximization. Since we were motivated by the suite of Pass@K-tailored algorithms, the set of algorithms we study here is broader. Additionally, beyond reverse-engineering, we put forth a specific forward-engineering recipe that forms the basis for using the framework to design new RLVR algorithms. That said, the expositions, even at the level of reverse-engineering, and the motivations between our paper and theirs are qualitatively different; we invite the interested reader to consult their paper as well for a complementary perspective on this topic.

The surrogate reward maximization objective in Eq. (3) falls under the umbrella of what the optimization literature calls *conditional stochastic optimization* (Hu et al., 2020a), which has been studied algorithmically in a series of works, e.g., (Hu et al., 2020b; Wang & Yang, 2022; He & Kasiviswanathan, 2023). The algorithm resulting from our forward-engineering is essentially the same as the biased stochastic gradient ascent algorithm proposed and studied in Hu et al. (2020b), but not identical: here we use the RLOO empirical estimator for $\nabla_\theta \rho_\theta$ rather than the REINFORCE estimator (no baseline), which would make the two identical. This connection of RLVR algorithms to the conditional stochastic optimization literature opens the possibility of applying techniques from it (e.g., variance reduction and debiasing) directly to RLVR implementations, and at the same time further motivates the study of the conditional stochastic optimization framework.

## 9 Conclusion

Motivated by an effort to reconcile recent Pass@K policy gradient methods, we show that advantage shaping is (asymptotically) equivalent to RLOO-style optimization of surrogate rewards $F(\rho)$. This equivalence extends beyond Pass@K: for instance, "hard-example upweighting" heuristics correspond to adding reward-level regularization $\Omega(\rho)$ to the training objective. While $F(\rho)$ is maximized at $\rho = 1$, acting as a data-fitting term, the regularizer $\Omega(\rho)$ provides an opposing force that can be interpreted as preventing overfitting, particularly to easy training examples.

Advantage shaping and surrogate reward design are two sides of the same coin. This equivalence mirrors classical supervised learning, where modifying the training objective is analogous to directly modifying the gradient. Both design approaches have proven effective, with a strong parallel in supervised classification for imbalanced data, e.g. (Lin et al., 2017; Menon et al., 2020; Ye et al., 2021; Kini et al., 2021). At a conceptual level, understanding this equivalence is instructive: irrespective of the design choice—whether starting from advantage shapes or surrogate rewards—analyzing the counterpart can provide valuable insight. Looking forward, while direct parallels to supervised learning should be drawn cautiously, the (regularized) surrogate reward lens may facilitate (i) a theoretical bridge between iterative policy-gradient optimization and the population performance metrics these algorithms ultimately aim to optimize, (ii) designing new practically relevant advantage shapes, with surrogate losses and reweighting schemes from supervised learning as potential starting points, and (iii) gaining insight into what properties (e.g., of the data distribution or model) determine when existing or new algorithms perform well.

## Acknowledgments

This work was supported by NSERC Discovery Grant No. 2021-03677, Alliance Grant ALLRP 581098-22, a gift from Google.org, and a UBC 4YF Departmental Fellowship. Additional computational resources were provided by the UBC Advanced Research Computing (ARC) Sockeye cluster. The authors thank the anonymous reviewers for the useful feedback that led to improving the manuscript. CT especially thanks Reinhard Heckel (TU Munich) and Mahdi Soltanolkotabi (USC) for inspiring and helpful discussions and ideas on the topic.

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

## Summary of Notations

**Notations.** For the reader's convenience, we list the core notation used throughout the paper.

$(x, a)$ — A problem-prompt $(x)$ and reference-answer $(a)$ pair.

$\pi_\theta(\cdot|x)$ — The policy (LLM) parameterized by $\theta$.

$N$ — Number of IID responses generated per example.

$y_i$ — The $i$-th response, $y_i \sim \pi_\theta(\cdot|x)$.

$r_i$ — The binary $0/1$ reward $r_{0/1}(y_i, a)$.

$\rho = \rho_\theta$ — The (population) expected $0/1$ reward $\mathbb{E}_{y \sim \pi_\theta(\cdot|x)}[r_{0/1}(y, a)]$.

$\widehat{\rho}$ — The empirical $0/1$ reward $\frac{1}{N} \sum_{i=1}^N r_i$.

$N^+, N^-$ — Number of correct $(r_i = 1)$ and wrong $(r_i = 0)$ responses.

$\rho_K = \rho_{K,\theta}$ — The (population) expected Pass@K reward.

$\widehat{\rho}_K$ — The unbiased empirical estimator of $\rho_{K,\theta}$.

$\nabla_i$ — The log-probability gradient $\nabla_\theta \log \pi_\theta(y_i|x)$.

$\widehat{\nabla}_+$ — Average empirical gradient over correct responses: $\frac{1}{N^+} \sum_{i:r_i=1} \nabla_i$.

$\widehat{\nabla}_-$ — Average empirical gradient over wrong responses: $\frac{1}{N^-} \sum_{i:r_i=0} \nabla_i$.

## A  GRPO's PPO-Style vs Surrogate Reward Objectives

In the original paper, Shao et al. (2024) present GRPO as optimizing a PPO-style objective (Schulman et al., 2017). This section reconciles that formulation with the surrogate reward objective derived in our Corollary 1.

**PPO-style objective.** Following common practice (Yu et al., 2025a; Liu et al., 2025; Zeng et al., 2025; Deng et al., 2025a), we omit the KL regularization term[9] and the $1/|y_i|$ length normalization (which is found responsible for length-bias) from the original GRPO objective of Shao et al. (2024), yielding:

$$\mathbb{E}_{\{y_i\}_{i \in [N]} \overset{\text{IID}}{\sim} \pi_{\theta_{\text{old}}}(\cdot|x)}\left[\frac{1}{N} \sum_{i=1}^N \sum_{t=1}^{|y_i|} \min(\gamma_{i,t} A_i, [\gamma_{i,t}]_{1-\epsilon}^{1+\epsilon} A_i)\right], \tag{33}$$

where $[\cdot]_a^b$ is the clip function at levels $a$ and $b$, and $\gamma_{it} = \pi_\theta(y_{it}|x, y_{i,<t})/\pi_{\theta_{\text{old}}}(y_{it}|x, y_{i,<t})$ is the policy ratio between the current model $\theta$ (to be updated) and the model $\theta_{\text{old}}$ that generates the responses. For clarity, further omit the clip operation, as done for example in Shao et al. (2024, App. A.1.6). Then the above objective simplifies to

$$\mathbb{E}_{y_i \sim \pi_{\theta_{\text{old}}}(\cdot|x)}\left[\frac{1}{N} \sum_{i=1}^N \sum_{t=1}^{|y_i|} \gamma_{i,t} \cdot A_i\right].$$

Note that the expectation here is defined with respect to samples from the *old* policy $\pi_{\theta_{\text{old}}}$, following the TRPO and PPO frameworks (Schulman et al., 2015; 2017). Thus, when taking gradients, $\pi_{\theta_{\text{old}}}$ is held fixed; only the numerator $\pi_\theta$ depends on $\theta$.

Taking the gradient with respect to $\theta$ and using the log-derivative trick, the gradient becomes

$$\mathbb{E}_{y_i \sim \pi_{\theta_{\text{old}}}(\cdot|x)}\left[\frac{1}{N} \sum_{i=1}^N \sum_{t=1}^{|y_i|} \gamma_{i,t} \cdot A_i \cdot \nabla_\theta \log \pi_\theta(y_{it}|x, y_{i,<t}|x)\right]. \tag{34}$$

---

[9]We refer the reader to Mroueh (2025) for a complementary analysis of GRPO's optimization dynamics focusing on the trajectory of the probability of success under explicit KL regularization towards a reference policy. Instead here, we focus solely on the implicit objective optimized by the gradients corresponding to the "advantage term" of the GRPO objective.

The fully online setting (corresponding to $\mu = 1$ iterations in (Shao et al., 2024, Alg. 1)) implies $\pi_\theta = \pi_{\theta_{\text{old}}}$. Then, by $\gamma_{i,t} = 1$, this simplifies to

$$\mathbb{E}_{y_i \sim \pi_\theta(\cdot|x)}\Big[\frac{1}{N}\sum_{i=1}^{N} A_i \cdot \nabla_\theta \log \pi_\theta(y_i|x)\Big]. \tag{35}$$

The empirical version of this, $\frac{1}{N}\sum_{i=1}^{N} A_i \cdot \nabla_\theta \log \pi_\theta(y_i|x)$, is precisely the GRPO gradient we analyze in Sec. 2 and Eq. (11).

**Difference to our surrogate reward.** A natural question arises: *How does the PPO-style objective in Eq. (34) relate to the surrogate reward $2\arcsin(\sqrt{\rho_\theta})$ that Corollary 1 identifies as the function optimized by the GRPO gradient (Eq. (35))?*

The resolution lies in recognizing that these are fundamentally different types of objectives:

- As mentioned, the PPO-style objective (Eq. (34)) is defined over a *fixed* distribution $\pi_{\theta_{\text{old}}}$.

- Our surrogate reward identifies the *on-policy* population objective that the gradient in Eq. (35) ascends.

To illustrate this distinction, consider what happens if we incorrectly interpret the PPO objective as fully on-policy by substituting $\pi_{\theta_{\text{old}}} = \pi_\theta$ everywhere (both in the integrand and the sampling distribution) in Eq. (34). This would yield:

$$\mathbb{E}_{y_i \sim \pi_\theta(\cdot|x)}\Big[\frac{1}{N}\sum_{i=1}^{N} A_i\Big]. \tag{36}$$

However, recall that $A_i = (r_i - \hat{\rho})/\sqrt{\hat{\rho}(1-\hat{\rho})}$: by construction, the advantages are zero-centered: $\sum_{i\in[N]} A_i = 0$ for any batch. Therefore, this naive on-policy interpretation yields a trivial objective that is identically zero; clearly not what GRPO optimizes.

As established in Corollary 1, the actual population objective that the on-policy GRPO gradient (Eq. (35)) ascends is the surrogate reward: $2\arcsin(\sqrt{\rho_\theta})$.

**Perspective Beyond the Fully Online Setting.** This surrogate reward perspective provides a principled way to interpret and design advantage functions. Our analysis shows how algorithms like GRPO and $\widetilde{\text{GRPO}}_K$ correspond to specific surrogate rewards in the on-policy setting. This link is constructive: new advantage-shaping methods, derived from new surrogate rewards (as in Sec. 3 and 6), are not limited to the fully online setting. Their resulting advantage scores, $A_i$, can be seamlessly embedded within the PPO-style objective (Eq. (33)), allowing them to be used in practical, multi-epoch training regimes.

# B  Missing Proofs

## B.1  Auxiliary Technical Results

**Lemma 1** (Relating $\widehat{f}_{K-1}^+$, $\widehat{f}_{K-1}^-$, and $\widehat{\rho}_K$). *With*

$$\widehat{\rho}_K = 1 - \frac{\binom{N^-}{K}}{\binom{N}{K}}, \qquad \widehat{f}_{K-1}^+ = \frac{\binom{N^-}{K-1}}{\binom{N-1}{K-1}}, \qquad \widehat{f}_{K-1}^- = \frac{\binom{N^--1}{K-1}}{\binom{N-1}{K-1}},$$

*and the conventions $\binom{N}{k} = 0$ when $N < k$ and $\binom{N}{0} = 1$, we have for $\widehat{\rho} = \frac{N^+}{N}$:*

$$1 - \widehat{\rho}_K = (1-\widehat{\rho}) \cdot \widehat{f}_{K-1}^- = (1-\widehat{\rho} - \frac{K-1}{N}) \cdot \widehat{f}_{K-1}^+ \quad .$$

*Proof.* Use the standard identities $\binom{N}{k} = \frac{N}{k}\binom{N-1}{k-1}$ and $\binom{N}{K} = \frac{N}{K}\binom{N-1}{K-1}$. Then

$$\frac{\binom{N^-}{K}}{\binom{N}{K}} = \frac{\frac{N^-}{K}\binom{N^--1}{K-1}}{\frac{N}{K}\binom{N-1}{K-1}} = \frac{N^-}{N} \cdot \frac{\binom{N^--1}{K-1}}{\binom{N-1}{K-1}} = \frac{N^-}{N}\,\widehat{f}^-_{K-1},$$

which implies $\widehat{\rho}_K = 1 - \frac{N^-}{N}\,\widehat{f}^-_{K-1}$.

Alternatively, using $\binom{N-1}{k} = \frac{N-k}{N}\binom{N}{k}$ with $N = N^-$ and $k = K-1$,

$$\widehat{f}^-_{K-1} = \frac{\binom{N^--1}{K-1}}{\binom{N-1}{K-1}} = \frac{N^- - (K-1)}{N^-} \cdot \frac{\binom{N^-}{K-1}}{\binom{N-1}{K-1}} = \frac{N^- - (K-1)}{N^-}\,\widehat{f}^+_{K-1}.$$

Substitute this into the previous display to get

$$\frac{\binom{N^-}{K}}{\binom{N}{K}} = \frac{N^-}{N} \cdot \frac{N^- - (K-1)}{N^-}\,\widehat{f}^+_{K-1} = \frac{N^- - (K-1)}{N}\,\widehat{f}^+_{K-1},$$

hence $\widehat{\rho}_K = 1 - \frac{N^- - (K-1)}{N}\,\widehat{f}^+_{K-1}$. The boundary conventions cover cases $K = 1$ (both weights = 1) and $N^- < K$ (ratio = 0). $\square$

**Lemma 2** (Conditional-gradient identity). *Let $r := r_{0/1}(y,a)$ and $\rho := \rho_\theta(x,a) := \mathbb{E}_{y \sim \pi_\theta(\cdot|x)}[r_{0/1}(y,a)]$. With expectations over $y \sim \pi_\theta(\cdot|x)$, define*

$$\Delta := \mathbb{E}[\nabla_\theta \log \pi_\theta(y|x) \mid r = 1] - \mathbb{E}[\nabla_\theta \log \pi_\theta(y|x) \mid r = 0].$$

*Then*

$$\Delta = \frac{1}{\rho(1-\rho)}\nabla_\theta\rho.$$

*Proof.* By conditioning and the log-derivative trick,

$$\mathbb{E}[\nabla_\theta \log \pi_\theta(y|x) \mid r = 1] = \frac{\mathbb{E}[r\,\nabla_\theta \log \pi_\theta(y|x)]}{\rho} = \frac{\nabla_\theta\rho}{\rho}.$$

Using $\mathbb{E}[\nabla_\theta \log \pi_\theta(y|x)] = 0$,

$$\mathbb{E}[\nabla_\theta \log \pi_\theta(y|x) \mid r = 0] = \frac{\mathbb{E}[(1-r)\,\nabla_\theta \log \pi_\theta(y|x)]}{1-\rho} = -\frac{\nabla_\theta\rho}{1-\rho}.$$

Subtracting gives $\Delta = \left(\frac{1}{\rho} + \frac{1}{1-\rho}\right)\nabla_\theta\rho = \frac{1}{\rho(1-\rho)}\nabla_\theta\rho$. $\square$

## B.2 Proof of Equation (15)

The leave-one-out unbiased estimator of Fail@(K-1) excluding response $i$ is:

$$1 - \widehat{\rho}^{\text{loo},i}_{K-1} = \frac{\binom{(N-1)(1-\widehat{\rho}^{\text{loo},i})}{K-1}}{\binom{N-1}{K-1}},$$

where the leave-one-out empirical 0/1 reward is

$$\widehat{\rho}^{\text{loo},i} = \begin{cases} \frac{N^+ - 1}{N-1} & \text{if } y_i \text{ is correct} \\ \frac{N^+}{N-1} & \text{if } y_i \text{ is wrong} \end{cases}.$$

For correct responses: $(N-1)(1 - \widehat{\rho}^{\text{loo},i}) = (N-1) - (N^+ - 1) = N^-$.

For wrong responses: $(N-1)(1 - \widehat{\rho}^{\text{loo},i}) = (N-1) - N^+ = N^- - 1$.

Apply now Lemma 1 to yield Eq. (15).

### B.3 Proof of Unbiasedness for RLOO$_K$

Fix an example $(x, a)$ and an index $i \in [N]$. Let $y^{\text{loo}_i} := \{y_j\}_{j \neq i}$ denote the other $N-1$ IID samples, and define for convenience the leave-one-out Fail@(K−1) weight $W^{\text{RLOO}_i} := 1 - \hat{\rho}_{K-1}^{\text{loo},i}$ and the RLOO baseline $b^{\text{loo}_i} := \frac{1}{N-1} \sum_{j \neq i} r_j$. As mentioned in the main body, the RLOO$_K$ gradient results from replacing in the gradient formula of $\widehat{G}_{\text{REINFORCE}_K}$ in Eq. (14) the reward $r_i$ with the vanilla RLOO advantage $A_i = r_i - b^{\text{loo}_i}$. Thus, the RLOO$_K$ estimator for sample $i$ is defined as:

$$\hat{g}_i := W^{\text{RLOO}_i} (r_i - b^{\text{loo}_i}) \nabla_i, \qquad \nabla_i := \nabla_\theta \log \pi_\theta(y_i|x).$$

Since $W^{\text{RLOO}_i}$ and $b^{\text{loo}_i}$ are functions of $y^{\text{loo}_i}$ only, they are independent of $(r_i, \nabla_i)$. Hence, conditioning on $y^{\text{loo}_i}$,

$$\mathbb{E}[\hat{g}_i] = \mathbb{E}\big[W^{\text{RLOO}_i} \mathbb{E}[(r_i - b^{\text{loo}_i})\nabla_i \mid y^{\text{loo}_i}]\big] = \mathbb{E}\big[W^{\text{RLOO}_i} \big(\mathbb{E}[r_i \nabla_i] - b^{\text{loo}_i}\mathbb{E}[\nabla_i]\big)\big]$$
$$= \mathbb{E}[W^{\text{RLOO}_i}] \cdot \mathbb{E}[r_i \nabla_i] = \big(1 - \rho_{K-1,\theta}(x,a)\big) \cdot \nabla_\theta \rho_\theta(x,a),$$

where we used $\mathbb{E}[\nabla_i] = 0$, $\mathbb{E}[r_i \nabla_i] = \nabla_\theta \rho_\theta(x,a)$, and $\mathbb{E}[W^{\text{RLOO}_i}] = 1 - \rho_{K-1,\theta}(x,a)$ since $W^{\text{RLOO}_i}$ is the standard unbiased estimator of Fail@(K−1) computed from $N-1$ samples. Averaging over $i$ and multiplying by $K$ yields $\mathbb{E}[\widehat{G}_{\text{RLOO}_K}(\theta; (x,a))] = K(1 - \rho_\theta)^{K-1}\nabla_\theta \rho_\theta = \nabla_\theta \rho_{K,\theta}$, proving unbiasedness.

### B.4 Proof of of Claim 1: Reverse-Engineering $\widetilde{\text{GRPO}}_K$

Recall the empirical update for the advantage-shaping method from Eq. (21):

$$\widehat{G}_{\widetilde{\text{GRPO}}_K} = \widetilde{\omega}_K \cdot \widehat{G}_{\text{GRPO}} = \sqrt{\frac{1 - \widehat{\rho}_K}{\widehat{\rho}_K}} \sqrt{\frac{\widehat{\rho}}{1 - \widehat{\rho}}} \cdot \widehat{G}_{\text{GRPO}}$$
$$= \sqrt{\frac{1 - \widehat{\rho}_K}{\widehat{\rho}_K}} \sqrt{\frac{\widehat{\rho}}{1 - \widehat{\rho}}} \cdot \sqrt{\widehat{\rho}(1 - \widehat{\rho})} \cdot \left(\widehat{\nabla}_+ - \widehat{\nabla}_-\right). \tag{37}$$

Replacing empirical $\hat{\cdot}$ quantities with their population counterparts, the population-level gradient (call it) $G_{\widetilde{\text{GRPO}}_K}$ is:

$$G_{\widetilde{\text{GRPO}}_K} = \left(\sqrt{\frac{1 - \rho_K}{\rho_K}} \cdot \sqrt{\frac{\rho}{1 - \rho}}\right) \sqrt{\rho(1 - \rho)} \cdot \mathbb{E}_{y_i \sim \pi_\theta(\cdot|x)}\left[\widehat{\nabla}_+ - \widehat{\nabla}_-\right].$$

The expectation in the last term becomes

$$\mathbb{E}_{y \sim \pi_\theta(\cdot|x)}[\nabla_\theta \log \pi_\theta(y|x) \mid r(y,a) = 1] - \mathbb{E}_{y \sim \pi_\theta(\cdot|x)}[\nabla_\theta \log \pi_\theta(y|x) \mid r(y,a) = 0].$$

Using Lemma 2, this equals $\nabla_\theta \rho / (\rho(1 - \rho))$. Thus,

$$G_{\widetilde{\text{GRPO}}_K} = \left(\sqrt{\frac{1 - \rho_K}{\rho_K}} \cdot \sqrt{\frac{\rho}{1 - \rho}}\right) \cdot \left(\frac{1}{\sqrt{\rho(1 - \rho)}} \nabla_\theta \rho\right) = \left(\sqrt{\frac{1 - \rho_K}{\rho_K}} \cdot \frac{1}{1 - \rho}\right) \cdot \nabla_\theta \rho. \tag{38}$$

We now relate $\rho$ to $\rho_K$. Recall that $\rho_K = 1 - (1 - \rho)^K \implies 1 - \rho = (1 - \rho_K)^{1/K}$. From this also $\nabla_\theta \rho_K = K(1 - \rho)^{K-1}\nabla_\theta \rho = K(1 - \rho_K)^{1 - 1/K}\nabla_\theta \rho$.

Substitute these two relationships back into (38):

$$G_{\widetilde{\text{GRPO}}_K} = \left(\frac{1}{K} \cdot \sqrt{\frac{1 - \rho_K}{\rho_K}} \cdot \frac{1}{(1 - \rho_K)^{1/K}(1 - \rho_K)^{1 - 1/K}}\right) \cdot \nabla_\theta \rho_K$$
$$= \left(\frac{1}{K\sqrt{\rho_K(1 - \rho_K)}}\right) \cdot \nabla_\theta \rho_K. \tag{39}$$

This is now in the form $G_{\widetilde{\text{GRPO}}_K} = F'(\rho_K)\nabla_\theta\rho_K$. We integrate in $\rho_K$ to find the surrogate $F$:

$$\int \frac{1}{\sqrt{\rho_K(1-\rho_K)}}\, d\rho_K = 2\arcsin(\sqrt{\rho_K}) + C = \arcsin(2\rho_K - 1) + C', \tag{40}$$

where $C, C'$ are constants. This completes the proof of the claim.

### B.5 Proof of Claim 4: Reverse-engineering Skew-R

Recall the empirical GRPO update (no clipping) from Eq. (11):

$$\widehat{G}_{\text{GRPO}} = \sqrt{\widehat{\rho}(1-\widehat{\rho})}\left[\widehat{\nabla}_+ - \widehat{\nabla}_-\right].$$

We have seen in the proof of Claim 1 that, at the population level this update becomes

$$G_{\text{GRPO}} = \frac{1}{\sqrt{\rho(1-\rho)}}\nabla_\theta\rho,$$

Then, the skew-R update at the population level is

$$G_{\text{skew-R}} = (1-\rho)\, G_{\text{GRPO}} = \sqrt{\frac{1-\rho}{\rho}}\, \nabla_\theta\rho.$$

We can more conveniently express this as

$$G_{\text{skew-R}} = \frac{1}{2\sqrt{\rho(1-\rho)}} \cdot \nabla_\theta\rho + \frac{1-2\rho}{2\sqrt{\rho(1-\rho)}} \cdot \nabla_\theta\rho\,.$$

We know from Corollary 1 that the first term corresponds to a surrogate reward $\arcsin(\sqrt{\rho})$. Analogously, the second term corresponds to $\Omega(\rho)$ for $\Omega$ such that $\Omega'(\rho) = \frac{1-2\rho}{2\sqrt{\rho(1-\rho)}}$. Integrating gives (up to a constant) $\Omega(\rho) = \sqrt{\rho(1-\rho)}$ completing the proof of the claim.

### B.6 Reward-Level Entropy Regularization

We start by computing the population gradient of the per example entropy-surrogate reward $2\arcsin(\sqrt{\rho}) + \lambda \cdot H(\rho)$, where recall $H(\rho) = -\rho\log(\rho) - (1-\rho)\log(1-\rho)$. By direct differentiation

$$G_{\text{entropy}}(\theta; (x,a)) = \left[\frac{1}{\sqrt{\rho(1-\rho)}} + \lambda \cdot \log\left(\frac{1-\rho}{\rho}\right)\right] \cdot \nabla_\theta\rho$$

Following the recipe of transforming the population objective into an empirical estimate we (1) substitute population quantities with their empirical counterparts in the multiplicative factor of the gradient, (2) substitute the population gradient with the an RLOO empirical gradient. We then get

$$\widehat{G}_{\text{entropy}}(\theta; (x,a)) = \left[\frac{1}{\sqrt{\widehat{\rho}(1-\widehat{\rho})}} + \lambda \cdot \log\left(\frac{1-\widehat{\rho}}{\widehat{\rho}}\right)\right] \cdot \widehat{\rho}(1-\widehat{\rho})\left[\widehat{\nabla}_+ - \widehat{\nabla}_-\right].$$

Recalling that $\widehat{G}_{\text{GRPO}} = \sqrt{\widehat{\rho}(1-\widehat{\rho})}\left[\widehat{\nabla}_+ - \widehat{\nabla}_-\right]$, we find

$$\widehat{G}_{\text{entropy}}(\theta; (x,a)) = \left[1 + \lambda \cdot \log\left(\frac{1-\widehat{\rho}}{\widehat{\rho}}\right) \cdot \sqrt{\widehat{\rho}(1-\widehat{\rho})}\right] \cdot \widehat{G}_{\text{GRPO}}\,.$$

Equivalently, the advantage scores of the new method are

$$A^\pm_{\text{entropy}} = \left[1 + \lambda \cdot \log\left(\frac{1-\widehat{\rho}}{\widehat{\rho}}\right) \cdot \sqrt{\widehat{\rho}(1-\widehat{\rho})}\right] \cdot A^\pm\,.$$

# C   Additional Analysis and Discussion of GRPO$_K$

## C.1   Reverse-Engineering GRPO$_K$

Taking limit of $N \to \infty$ and replacing empirical quantities with population counterparts in Eq. (18), we find

$$G_{\mathrm{GRPO}_K} = \frac{1 - \rho_K}{1 - \rho} \sqrt{\rho(1-\rho)} \cdot \Delta$$

where the term $\Delta = \nabla_\theta \rho / (\rho(1-\rho))$ is the difference of correct/wrong population gradients defined in Lemma 2. Simplifying,

$$G_{\mathrm{GRPO}_K} = \frac{(1-\rho)^{K-1}}{\sqrt{\rho(1-\rho)}} \cdot \nabla_\theta \rho \,.$$

Integrating the multiplicative factor $(1-\rho)^{K-\frac{3}{2}}\rho^{-\frac{1}{2}}$ over $\rho$ the per-example surrogate reward that gives this gradient is the incomplete beta function with parameters $1/2$ and $K - 1/2$:

$$\mathcal{B}\big(\rho; 1/2, K - 1/2\big) \,.$$

Written in terms of Pass@K reward, the surrogate reward is

$$\mathcal{B}\big(1 - (1 - \rho_K)^{1/K}; 1/2, K - 1/2\big) \,. \tag{41}$$

By symmetry properties of the incomplete beta function, this is up to (a $K$-dependent) constant equivalent to $-\mathcal{B}\big((1 - \rho_K)^{1/K}; 1/2, K - 1/2\big)$.

For $K = 1$,

$$\mathcal{B}\big(1 - (1 - \rho_K)^{1/K}; 1/2, K - 1/2\big) = \mathcal{B}\big(\rho_K; 1/2, 1/2\big) = \int_0^{\rho_K} \frac{1}{\sqrt{\tau(1-\tau)}} \mathrm{d}\tau = 2\arcsin(\sqrt{\rho_K}) \,.$$

Thus, the surrogate function coincides with that of $\widetilde{\mathrm{GRPO}}_K$ (Claim 2). However, for $K \geq 2$, the two surrogates differ. Specifically, GRPO$_K$'s surrogate is adaptive to the value of $K$. See Fig. 5.

## C.2   GRPO$_{K=2}$ vs. Skew-R

For $K = 2$: $\widehat{f}_{K-1}^+ = (1 - \widehat{\rho}_2 - 1/N)/(1 - \widehat{\rho})$ and $\widehat{f}_{K-1}^- = (1 - \widehat{\rho}_2)/(1 - \widehat{\rho})$, which applied to Eq. (18) gives

$$\widehat{G}_{\mathrm{GRPO}_{K=2}} = \frac{1 - \widehat{\rho}_2 - 1/N}{1 - \widehat{\rho}} \sqrt{\widehat{\rho}(1-\widehat{\rho})} \cdot \widehat{\nabla}_+ - \frac{1 - \widehat{\rho}_2}{1 - \widehat{\rho}} \sqrt{\widehat{\rho}(1-\widehat{\rho})} \cdot \widehat{\nabla}_- \,.$$

For $N \gg 1$, for which $1 - \widehat{\rho}_2 - 1/N \approx 1 - \widehat{\rho}_2 \approx (1 - \widehat{\rho})^2$, this becomes the same as the skew-R objective in Eq. (31): $(1 - \widehat{\rho})\sqrt{\widehat{\rho}(1-\widehat{\rho})} \big[\widehat{\nabla}_+ - \widehat{\nabla}_-\big]$. This raises a question with regards to our reverse-engineering perspective: On the one hand, we have argued that at population level skew-R maximizes $\arcsin(\sqrt{\rho}) + \sqrt{\rho(1-\rho)}$. On the other hand, we have seen in the previous subsection that GRPO$_{K=2}$ maximizes $\mathcal{B}\big(1 - (1 - \rho_2)^{1/2}; 1/2, 3/2\big)$.

First, note that the two surrogates are in fact algebraically identical. To see this, we can express $\mathcal{B}\big(1 - (1 - \rho_2)^{1/2}; 1/2, 3/2\big)$ in terms of $\rho$ (using the definition $\rho_2 = 1 - (1 - \rho)^2$):

$$\mathcal{B}\big(\rho; 1/2, 3/2\big) = \int_0^\rho \sqrt{\frac{1 - \tau}{\tau}} \mathrm{d}\tau = \int_0^\rho \frac{1}{2\sqrt{\tau(1-\tau)}} \mathrm{d}\tau + \int_0^\rho \frac{1 - 2\tau}{2\sqrt{\tau(1-\tau)}} \mathrm{d}\tau \,.$$

The first integral is $\arcsin(\sqrt{\rho})$ and the second is $\sqrt{\rho(1-\rho)}$, confirming the identity. This means there is no inconsistency in our reverse-engineering strategy. At the same time, the simple three-step forward-engineering recipe operates by substituting population quantities with empirical estimates without accounting for unbiasedness (e.g., via leave-one-out): thus it cannot recover the finite-sample ($N$ not $\gg 1$) version of GRPO$_K$ and arrives at skew-R.

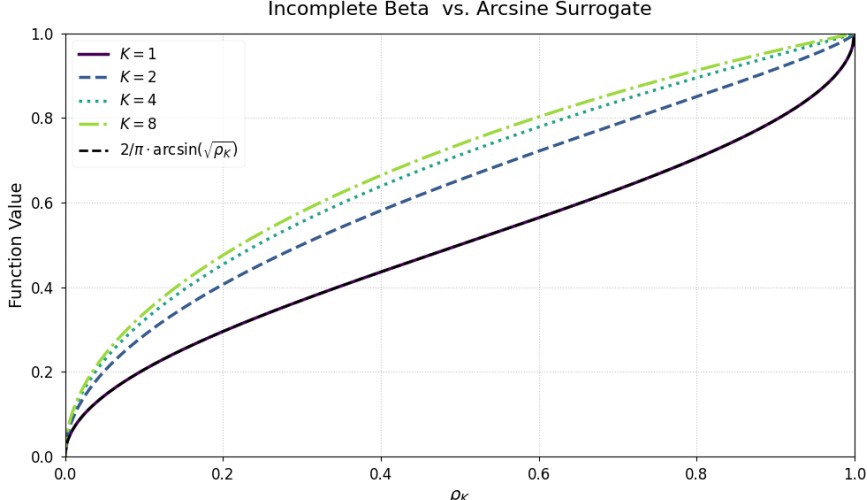

Figure 5: A comparison of surrogate reward functions, plotted against the Pass@K reward $\rho_K$. The colored lines show the $\mathrm{GRPO}_K$ surrogate, Eq. (41) for various $K$ values. The black dashed line shows the (normalized) $\widetilde{\mathrm{GRPO}}_K$ surrogate, $\frac{2}{\pi}\arcsin(\sqrt{\rho_K})$. (All curves are normalized to reach value 1 at $\rho_K = 1$.) The surrogates agree for $K = 1$ (dashed and purple solid lines coincide), but for $K > 1$ the $\mathrm{GRPO}_K$ surrogates become concave providing stronger optimization incentive for hard prompts.

This algebraic identity, however, raises a subtle question of interpretation for how we talk about "regularization." One reading (a *0/1-reward-centric* lens) is to say: $\mathrm{GRPO}_{K=1}$ already optimizes the surrogate $\arcsin(\sqrt{\rho})$; moving to $K = 2$ simply adds the extra term $\sqrt{\rho(1-\rho)}$, which one might then call a reward-level regularizer

However, for the Pass@K methods, we argue it is more appropriate to adopt a *Pass@K-centric* lens. For Pass@K methods, $\rho_K$ is the actual performance metric of interest, and the 0/1 reward ($\rho$) is merely an intermediate quantity used in its calculation (since $\rho_K = 1 - (1-\rho)^K$). From this perspective, the objective in Eq. (41) is not a 'regularized 0/1 surrogate' but rather a *direct surrogate for $\rho_K$*: a monotone transformation of the Pass@K objective itself. Under this view, $\mathrm{GRPO}_K$ is just doing standard surrogate maximization of the performance metric (analogous to cross-entropy as a surrogate for 0/1 risk), rather than "optimizing $\rho$ with an added regularizer."

Both interpretations are mathematically consistent; the choice depends on whether one views the optimization goal as improving the 0/1 reward (and thus interpreting $K > 1$ as regularization) or as directly optimizing the Pass@K metric (and thus interpreting the objective as a simple, monotone transformation of $\rho_K$).

## C.3 Comparison to Biased Scaling

In Sec. 3 we applied the log-derivative trick directly to the Pass@K objective, yielding the expected gradient $G_{\mathrm{Pass@K}}(\theta; (x, a))$ in Eq. (12). We then derived Pass@K variants of REINFORCE by using a leave-one-out construction to obtain unbiased estimates of the two components in (12): Fail@$(K-1)$ and $G_{0/1}(\theta; (x, a))$, resulting in the unbiased empirical gradient $\widehat{G}_{\mathrm{RLOO}_K}$ in Eq. (17).

In contrast, Mahdavi et al. (2025) apply a *biased* empirical gradient that estimates the Fail@$(K-1)$ factor as $(1 - \widehat{\rho})^{K-1}$. The corresponding advantage-score scaling is (up to constant multiplicative factor $K$)

$$(1 - \widehat{\rho})^{K-1} \cdot A^{\pm}. \tag{42}$$

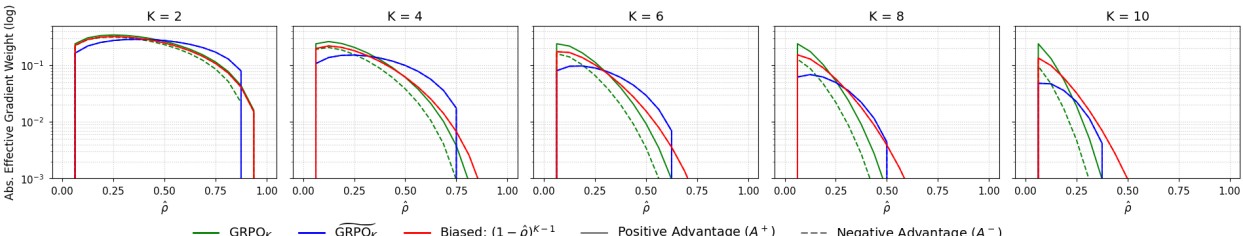

Figure 6: Comparison of *absolute* effective gradient weights (log scale) for Pass@K training with $N = 16$. Each panel shows a different $K$ and plots the magnitude applied to correct (solid) and wrong (dashed) responses under three schemes: $\text{GRPO}_K$ (Eq. (19); green), $\widetilde{\text{GRPO}_K}$ (Eq. (21); blue), and the biased scaler (Eq. (42); red). For small $\hat{\rho}$, $\text{GRPO}_K$ assigns slightly larger weights than the biased scaler. As $\hat{\rho}$ increases, $\text{GRPO}_K$ enforces a hard zero once $\hat{\rho} > 1 - \frac{K-1}{N}$, whereas the biased scaler decays smoothly. $\widetilde{\text{GRPO}_K}$ applies a symmetric, example-level scale and often yields the largest mid-$\hat{\rho}$ weights until $\hat{\rho}_K$ saturates. Only the $\text{GRPO}_K$ exhibits asymmetric scaling between correct and wrong responses.

Unlike the estimators in Sec. 3, this remains biased even if the advantage-scores $A^{\pm}$ correspond to an unbiased method (e.g., RLOO), because $(1 - \hat{\rho})^{K-1}$ is itself a biased estimator of $\text{Fail@}(K - 1)$. Replacing it by a leave-one-out version $(1 - \hat{\rho}^{\text{loo}})^{K-1}$ does not remove this bias.

Note that such biased scaling is also present in advantage-shapes resulting from our forward-engineering of surrogate rewards in Sec. 5. Even our reverse-engineering operates at the population level. Thus, here, it would tell us that both $\text{GRPO}_K$ (which applies unbiased reweightings to vanilla GRPO) and the method of Mahdavi et al. (2025), both implicitly maximize the same surrogate reward (as derived in App. C.1). In the practical regime of few samples though the two methods could lead to different behaviors. While unbiasedness is intuitively a good property, we discuss below regimes where the biased variant can be particularly attractive. It is worth noting that for $K = 2$ the biased variant coincides with skew-R.

**When a biased reweighting factor can help.** There are practical regimes where biased scalings are intuitively attractive. Specifically, for Pass@K optimization this includes (see also Fig. 6): *(1) Compute-constrained $N \leq K$.* The unbiased Pass@K estimator in Sec. 3 (and the grouping-based method of Sec. 4) is undefined when $K > N$, whereas $(1 - \hat{\rho})^{K-1}$ still yields a meaningful signal. *(2) Small N relative to K.* When $N > K$ but not by much, the unbiased combinatorial weight becomes exactly zero whenever $N^{-} < K - 1$, prematurely killing gradients; the smooth scaler $(1 - \hat{\rho})^{K-1}$ avoids this early saturation and provides a gentler shaping $A_{\text{Pass@K}}^{\pm} = (1 - \hat{\rho})^{K-1} A^{\pm}$ that maintains nonzero updates. *(3) Very large N relative to K.* When $N \gg K$, $\hat{f}_{K-1}^{+} \approx \hat{f}_{K-1}^{-} \approx \left(N^{-}/(N - 1)\right)^{K-1} \approx (1 - \hat{\rho})^{K-1}$, so the two estimators nearly coincide.

## C.4 Comparison to Further Related Work

In the main body, we compared $\text{GRPO}_K$ to its biased variant from Mahdavi et al. (2025) and to $\widetilde{\text{GRPO}_K}$ as implemented by Chen et al. (2025). Policy optimization for Pass@K has also been studied by Walder & Karkhanis (2025). Their approach differs from both ours and Mahdavi et al. (2025) in two important ways.

First, Walder & Karkhanis (2025) do not formulate GRPO-style estimators. The fair comparison is therefore with our $\text{REINFORCE}_K$ / $\text{RLOO}_K$ estimators from Sec. 3, which directly target the Pass@K objective. Like us, they construct an unbiased estimator of $\nabla_\theta \rho_{K,\theta}(x, a)$, but the structure of their estimator is different.

In our notation (and dropping an overall prefactor $K$ that is common across examples and matches Sec. 3), their per-example estimator for $\nabla_\theta \rho_{K,\theta}(x,a)$ has the following form:

$$\frac{1}{N} \sum_{i \in [N]} \nabla_\theta \log \pi_\theta(y_i|x) \cdot \begin{cases} 1, & \text{if } r_{0/1}(y_i, a) = 1, \\ 1 - \binom{N^- - 1}{K-1} / \binom{N-1}{K-1}, & \text{if } r_{0/1}(y_i, a) = 0. \end{cases} \tag{43}$$

This should be contrasted with our $\text{REINFORCE}_K$ estimator which can be interpreted as a reweighting of vanilla REINFORCE. By contrast, the estimator in Eq. (43) assigns a *positive* coefficient not only to correct responses but also to *incorrect* responses. That is, it explicitly takes *positive* gradient steps in directions corresponding to some incorrect samples in the batch. Thus, it is also not comparable to $\text{RLOO}_K$.

Besides, it is not immediately clear from (43) how to subtract a baseline to reduce variance while maintaining unbiasedness. Instead, going from $\text{REINFORCE}_K$ to $\text{RLOO}_K$ in Sec. 3 was rather intuitive. We refer the reader to (Walder & Karkhanis, 2025, Sec. 4) for more discussion on variance reduction of their method.

For completeness, we can apply our reverse-engineering lens to Eq. (43) and verify that it indeed ascends the Pass@K objective in expectation. Rewriting Eq. (43) more compactly using our notation for conditional gradients $\widehat{\nabla}_+$ and $\widehat{\nabla}_-$ and for $\widehat{f}_{K-1}^-$, we obtain (for a single $(x,a)$)

$$\hat{\rho}\,\widehat{\nabla}_+ \;+\; (1-\hat{\rho})\left(1 - \widehat{f}_{K-1}^-\right)\widehat{\nabla}_- \,.$$

Passing to population quantities (Lemmas 1 and 2) and using $\rho_K = 1 - (1-\rho)^K$, this expression becomes

$$\rho \cdot \frac{\nabla_\theta \rho}{\rho} \;-\; (1-\rho)\left(1 - \frac{1-\rho_K}{1-\rho}\right) \cdot \frac{\nabla_\theta \rho}{1-\rho} \;=\; \left(\frac{1-\rho_K}{1-\rho}\right)\nabla_\theta \rho \;=\; (1-\rho)^{K-1}\,\nabla_\theta \rho \;=\; \frac{1}{K}\,\nabla_\theta \rho_{K,\theta}.$$

Thus, in population, the estimator from Walder & Karkhanis (2025) is indeed maximizing $\rho_{K,\theta}$, consistent with our own derivations. The difference is at finite-samples: (i) how gradient weight is assigned across samples (especially incorrect ones), and (ii) how variance reduction is handled at finite $N$.

Finally, Walder & Karkhanis (2025) explicitly emphasize extending Pass@K-style optimization to non-binary continuous rewards. It would be interesting future work to extend the surrogate reward lens to such reward settings beyond binary.

## D  Empirical validations

Here we provide *preliminary* empirical validation of several policy-gradient methods discussed in this paper on controlled mathematical reasoning tasks. The goal of these experiments is not a comprehensive benchmark, but rather to demonstrate that the derived/identified updates can be implemented and can be competitive with the strong GRPO baseline in a controlled setting. As discussed in Sec. 7, a systematic empirical study across tasks, compute regimes, and hyperparameter choices is beyond the scope of this paper and is left to future work.

### D.1  Synthetic Data

**Setting.** We perform a lightweight round of supervised fine-tuning (SFT) on Qwen2.5-0.5B-Base Yang et al. (2024) using 1,000 randomly sampled problems from OpenMathInstruct-2 (Toshniwal et al., 2024). This SFTed model serves as the baseline and primarily learns proper formatting and instruction-following behavior for mathematical problems.

We then apply RL to a synthetic arithmetic task of the following form: `What is` $a \times b + c \times d$`?` The training, validation, and test splits consist of 50,000, 500, and 500 problems, respectively. For training, we use a learning rate of $10^{-6}$, weight decay 0.01, sampling temperature 1, 16 unique prompts per optimization step, and $N = 8$ rollouts per prompt (i.e., batch size of 128 sequences per optimization step). The training is done on-policy for eight epochs, and without KL regularization term.

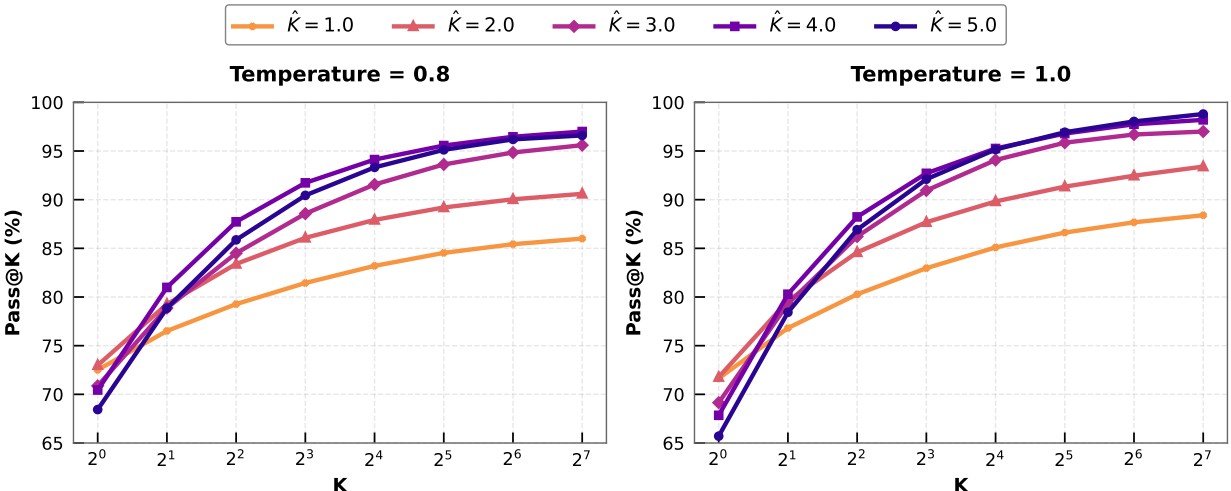

Figure 7: Performance of (biased) $\text{GRPO}_{\hat{K}}$ (Eq. (42) with $K \leftarrow \hat{K}$). $\hat{K} = 1$ corresponds to vanilla GRPO, while $\hat{K} = 2$ is equivalent to skew-R. Larger values of $\hat{K}$ explicitly target higher Pass@K. Empirically, increasing $\hat{K}$ improves Pass@K at larger evaluation values of $K$, but degrades performance at smaller $K$. Gains diminish as $\hat{K}$ increases.

To avoid ambiguity between the Pass@K metric used for evaluation and the targeted Pass@K objective used during training, we denote $\hat{K}$ as the target value of $K$ used for training, and $K$ as the value used for evaluation. For evaluation we use a test-time rollout size of $M = 128$.

We evaluate the following algorithms: $\text{GRPO}_K$ (Eq. (18)), the biased $\text{GRPO}_K$ implementation of Mahdavi et al. (2025) (Eq. (42)), $\widetilde{\text{GRPO}}_K$ of Chen et al. (2025) (Eq. (22), our skew-R variant (Eq. (31)), and the entropy-regularized method (Eq. (32)). For unbiased and biased $\text{GRPO}_K$ and $\widetilde{\text{GRPO}}_K$, we set $K \leftarrow \hat{K}$ (with $\hat{K} \in \{2, 3\}$) in Eqs. (18), (42), and (22) respectively. For the entropy method, we set $\lambda = 1$.

**Higher values of $\hat{K}$ lead to higher Pass@K.** Figure 7 shows the effect of varying the target $\hat{K}$ during training on downstream Pass@K performance. We observe that increasing $\hat{K}$ generally improves Pass@K at larger evaluation values of $K$. However, this improvement comes at the cost of reduced performance for smaller $K$, suggesting a trade-off between single-sample accuracy and multi-sample coverage. Moreover, the gains diminish as $\hat{K}$ increases, indicating diminishing returns for very large target values.

**Algorithm comparisons.** Figure 8 compares the performance of the algorithms across values of $K$ used for evaluation and for sampling temperatures 0.8 and 1. Recall that for $\hat{K} = 2$, the biased $\text{GRPO}_K$ variant is identical to skew-R.

For $\hat{K} = 2$ (Figure 8a), the unbiased estimator outperforms the biased estimator (which in this case coincides with skew-R) across most evaluation values of $K$, suggesting that reducing finite-$N$ bias can be beneficial when optimizing for relatively small target $K$. Recall from Sec. C.2 that for $\hat{K} = 2$ skew-R and the biased estimator correspond to the same *population* surrogate objective, which does not capture the finite-$N$ effects reflected in this experiment.

In contrast, for $\hat{K} = 3$ (Fig. 8b), the biased estimator achieves better performance at larger evaluation values of $K$, indicating that the finite-sample trade-off can shift with $\hat{K}$. This may be due to the higher variance of the unbiased estimator in this regime, or because the unbiased estimator too aggressively dampens the advantage scores for high-$\hat{\rho}$ prompts, as seen in Fig. 2.

Across both settings: the entropy-based method achieves stronger Pass@1, indicating improved single-sample accuracy. However, as expected, at higher Pass@K values, it underperforms both the biased and unbiased

$\text{GRPO}_K$ estimators which aim to directly maximize a surrogate of the Pass@K reward $\rho_{K,\theta}$. Furthermore, $\widetilde{\text{GRPO}}_K$ performs on par with the biased $\text{GRPO}_K$ estimator.

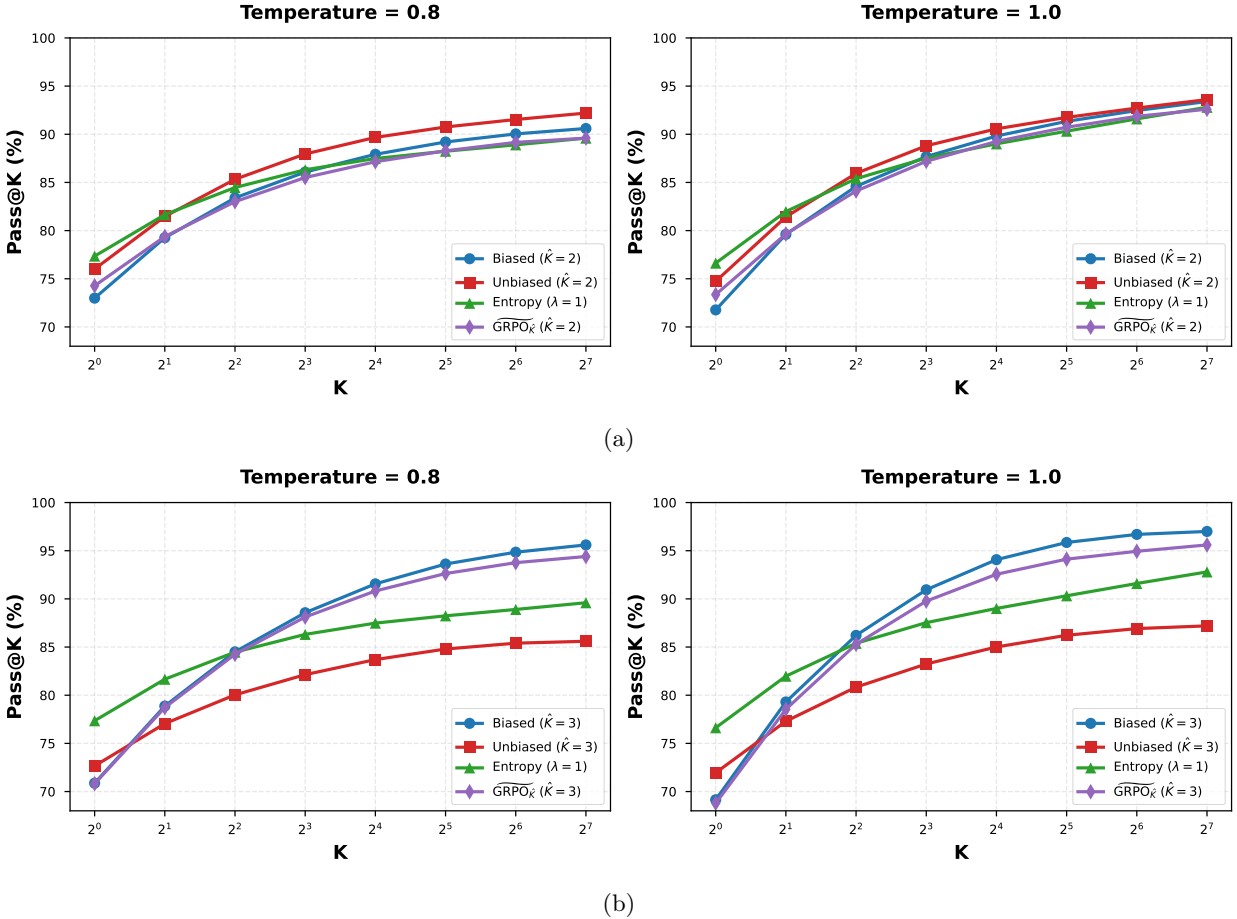

Figure 8: Performance comparison of different methods on the synthetic dataset of Sec. D.1, using $N = 8$ responses per prompt and test-time sampling temperature 0.8 (Left) and 1 (Right). (a) $\hat{K} = 2$: The unbiased $\text{GRPO}_{\hat{K}}$ is best performing across $K \geq 2$. The entropy-based method achieves stronger 0/1 reward (i.e., Pass@1) (b) $\hat{K} = 3$: The biased $\text{GRPO}_{\hat{K}}$ is best performing for $K \geq 4$. $\widetilde{\text{GRPO}}_K$'s performance is comparable. The entropy-based method yields again the strongest Pass@1 performance.

## D.2 Real math problems

Here, we conduct additional initial evaluations on real-world mathematical reasoning tasks. Specifically, we compare three methods: GRPO, Pass@K-mixed (Eq. (29), $\hat{K} = 4$), and Skew-R (Eq. (31)). Following the training protocol of Deng et al. (2025b), we fine-tune Qwen2.5-Math-1.5B (Yang et al., 2024) on the MATH dataset (levels 3–5) (Hendrycks et al., 2021). To accelerate training, we adopt dynamic sampling (Yu et al., 2025b), which discards samples with zero advantage and resamples until a full batch is formed. All methods share identical reinforcement-learning hyperparameters. Concretely, training is performed on four A100 GPUs with a prompt batch size of 256 and $N = 8$ rollouts per prompt. We use a learning rate of $1 \times 10^{-6}$ and a mini-batch size of 64, yielding 32 gradient updates per step. Training runs for 40 steps, corresponding to more than two effective epochs due to the increased throughput from dynamic sampling. We set the sampling temperature to 1.0, the clipping ratio to 0.2, and the KL regularization coefficient to $1 \times 10^{-4}$. We evaluate on the challenging AIME2024, AIME2025, and AMC23 benchmarks using unbiased

| Method | Qwen2.5-Math-1.5B Pass@K | | | | | | | | |
|---|---|---|---|---|---|---|---|---|---|
| | 1 | 2 | 4 | 8 | 16 | 32 | 64 | 128 | 256 |
| **AIME 2025** | | | | | | | | | |
| GRPO | 5.9 | 9.9 | 15.0 | 20.5 | 26.5 | 33.6 | 41.5 | 49.8 | 56.7 |
| Pass@K-mixed | 5.6 | 9.6 | 14.6 | 20.1 | 26.1 | 33.3 | 41.7 | 50.0 | 56.7 |
| skew-R | 5.7 | 9.7 | 14.9 | 20.4 | 25.9 | 31.8 | 37.8 | 44.8 | 53.3 |
| **AIME 2024** | | | | | | | | | |
| GRPO | 11.4 | 17.7 | 24.3 | 30.5 | 36.7 | 43.4 | 50.0 | 56.0 | 63.3 |
| Pass@K-mixed | 10.6 | 16.7 | 23.5 | 30.3 | 37.1 | 44.3 | 51.2 | 57.5 | 63.3 |
| skew-R | 10.2 | 16.0 | 22.3 | 28.3 | 34.7 | 42.3 | 50.8 | 60.3 | 70.0 |
| **AMC23** | | | | | | | | | |
| GRPO | 46.6 | 59.1 | 70.0 | 78.9 | 85.5 | 90.2 | 93.7 | 96.0 | 97.5 |
| Pass@K-mixed | 45.2 | 58.1 | 69.4 | 78.4 | 85.3 | 90.2 | 95.2 | 98.5 | 100.0 |
| skew-R | 45.7 | 58.3 | 69.0 | 77.6 | 84.5 | 90.0 | 94.8 | 98.6 | 100.0 |
| **Average** | | | | | | | | | |
| GRPO | **21.3** | **28.9** | **36.4** | **43.3** | **49.6** | 55.7 | 61.7 | 67.3 | 72.5 |
| Pass@K-mixed | 20.5 | 28.1 | 35.8 | 42.9 | 49.5 | **56.1** | **62.7** | **68.7** | 73.3 |
| skew-R | 20.5 | 28.0 | 35.4 | 42.1 | 48.4 | 54.7 | 61.1 | 67.9 | **74.4** |

Table 2: Performance comparison of different methods (see text) on AIME 2025, AIME 2024, and AMC23 under Qwen2.5-Math-1.5B.

Pass@$K$ accuracy with temperature 1.0, which better reflects exploration settings. Final performance is reported using the unbiased Pass@K metric with a test-time rollout size of $M = 256$.

Table 2 shows that Skew-R achieves strong average performance at $K = 128$ and $K = 256$, matching or surpassing pass@K-mixed. Since large-$K$ primarily reflects the ability to discover diverse correct solutions rather than improving single-sample accuracy, these gains suggest that Skew-R enhances exploration. This observation supports our interpretation that reallocating gradient emphasis toward low-success prompts leads to broader solution coverage. We expect these effects to be amplified when the rollout-size $N$ is larger; such ablations are out of scope and we leave them to future work.

