# OpenReview forum: "Advantage Shaping as Surrogate Reward Maximization: Unifying Pass@K Policy Gradients"
_TMLR — Accepted by TMLR_

### Review · Reviewer_Rab9 · 2025-12-18

**Summary Of Contributions:**

This paper provides a theoretical bridge between Reinforce-style gradients to GRPO for optimizing the Pass@K metric in LLMs.

*Strength*

1. Method is indeed novel and theoretical grounded.
2. Applied to different methods from reinforce, rloo to GRPO.

*Weaknesses*

1. Some derivation isn't rigor. See my detailed comment below.
2. Presentation issue exists. See my detailed comment below.
3. Pass@k reward has already been utilized RLVR, but this work didn't justify the effectiveness of the proposed method in actual training.

**Additional Comments:**

There are some parts that I didn't yet follow and may need author's clarification to further carefully check the full proof, will update by adding new thread of comment if I have any concerns or questions.

**Audience:**

Yes

**Audience Explanation:**

Yes.

LLM RLVR with pass@K reward is a timely topic, and many readers will be benefited from the theoretical analysis in this paper.

**Claims And Evidence:**

Yes

**Claims Explanation:**

To some extent, yes.

While the paper provides a compelling framework for unifying Pass@k objectives, the central claims of theoretical equivalence are occasionally undermined by heuristic derivation steps and notation inconsistencies that introduce statistical bias not fully resolved in the proofs.

**Requested Changes:**

1. Writing issues, please ensure rigor as this seems to be a more theoretical work
- Please correct all the citation format, i.e., using \citep and \citet properly, currently one affects reading.
- The definition of Pass@K sampling isn't clear in terms of RLVR practice (page 2 below section 1.1.2); it would be better to define sth like "independently sampling K response in xxx temperature from prompt x". The current explanation of repetitive prompting reads less rigorous.
- Please rigorously define all notations at the beginning of the main-body work (move the one from appx to main body) and double check all the expressions and phrases to make sure they are **consistent**. What do you mean by "per-example" expected pass@k reward, as $x$ is defined as "problem" instead of "example".
- I don't quite follow the derivation in eq23 to eq24, somehow confused by $\hat{f}_{K-1}$ and $\omega$.

2. Theoretical rigor
- In eq6, there's a sample-level coefficient $\frac{N}{N-1}$ and you say you can "ignore it" below the line under eq6, but I don't quite understand why this is removed in GRPO eq8? Simply dropping it, the estimator then becomes biased by a constant multiplicative factor.
- For eq15, those quantities are over batch, which $\hat f$ and $\hat \nabla$ are calculated over the same batch. How does this ensure the unbiasedness?
- Sec5.2 claim 2 might have some issue. Could you please clarify the following questions: If I understand correctly, the derivation relies on substituting population quantities ($F'(\rho)$ and $\nabla \rho$) with empirical, batch-dependent estimators ($F'(\hat{\rho})$ and $\hat{G}_{RLOO}$). Since these two terms are computed from the same sample batch, they actually are statistically dependent, so it yields a biased estimation, which seems to be a heuristic approximation? How it makes sure that this batch-dependency with RLOO estimator to maintain independence between the weight and the sample $y_i$? If so, can the error towards population-level surrogate be bounded?
- GRPO has KL, clipping and importance ratio sampling. I didn't find any assumption or clarification regarding the GRPO objective that you are analyzed until page 12. This should be moved to the beginning to avoid misleading confusions.

3. Actual training effectiveness.

Could you provide some actual experiment by comparing to other pass@K RLVR method (e.g., SimKO)? The readers will be more interested in how the derived surrogate would work actual training. This is similar to how DPO simplifies from RLHF, the authors not only show the nice reparameterization of RLHF objective but also demonstrate it via actual experiment result.

---

> ### Author Response · Authors · 2026-01-21
>
> Thank you for your careful read, useful feedback and detailed questions. We have revised the document accordingly and further respond to your questions below.
>
> **citation format**
>
> Thank you. We have corrected this in the revision.
>
> **Define notations + What do you mean by "per-example" expected pass@k reward**
>
> We have expanded the notation paragraph in the main body to explicitly define the key notations up front.
> Regarding terminology: we use “per-example” because the expectation is taken conditional on a fixed example $(x,a)$ (i.e., a fixed prompt and reference answer); the reward depends on both $x$ (through the conditional policy $\pi_\theta(\cdot\mid x)$) and $a$ (through the verifier $r(\cdot,a)$).
>
> **I don't quite follow the derivation in eq23 to eq24**
>
> To clarify: Eq. (23) *does not derive* Eq. (24). They are two separate definitions of two different “mixing” schemes.
>
> * **Eq. (23)** restates the mixing rule proposed by Chen et al.: they mix **their** Pass@K-shaped advantage $\widetilde A^\pm_K$ with the vanilla GRPO advantage $A^\pm$ using $\hat\rho$ as the mixing coefficient. This is the **first equality** in Eq. (23). The **second equality** is just an algebraic simplification obtained by substituting the proportionality $\widetilde A^\pm_K = \widetilde\omega \, A^\pm$ from Eq. (19).
> * **Eq. (24)** defines an *analogous* mixing rule, but mixing vanilla GRPO advantage $A^\pm$ with **our** Pass@K-shaped advantage $A^\pm_K$. This is the **first equality** in Eq. (24). The **second equality** again follows by substituting $A^\pm_K = \omega_\pm \, A^\pm$ from Eq. (17).
>
> So, the only “steps” involved in the second equalities are substitutions of Eqs. (19) and (17), respectively; there is no derivation from Eq. (23) to Eq. (24).
>
> **why the N/(N-1) factor is removed in GRPO eq8?  the estimator then becomes biased**
>
> To clarify: we do not remove an $N/(N-1)$ factor in GRPO (Eq. (8)). The $N/(N-1)$ prefactor is specific to **RLOO** due to the leave-one-out baseline (Eq. (6)); GRPO uses the standardized advantage $A_i=(r_i-\hat\rho)/\sqrt{\hat\rho(1-\hat\rho)}$ without LOO estimates, so there is no such factor in the first place.
> Also, GRPO is biased regardless (due to the normalization by $\sqrt{\hat\rho(1-\hat\rho)}$), which we note explicitly in the main text (e.g., the “Biased vs unbiased estimation” discussion).
>
> **For eq15, those quantities are over batch, which $\rho$ and $\rho$ are calculated over the same batch. How does this ensure the unbiasedness**
>
> Thank you for the careful read. This is indeed a subtle point: in general the product of two unbiased batch estimators can be biased due to dependence; however, here this does not occur because we use a *leave-one-out* estimator of the Fail@(K-1) factor (excluding the same sample whose log-gradient appears in the term).
>
> Here, Eq. (15) should not be read as “a batch estimate of Fail@(K-1) multiplied by a batch gradient.” The unbiased estimator is the *per-sample leave-one-out* form in Eq. (12) (and its RLOO variant obtained by replacing $r_i$ with a baseline that excludes sample $i$). The leave-one-out Fail@(K-1) factor $1-\widehat\rho_{K-1}^{\text{loo},i}$ depends only on $\{y_j\}_{j\neq i}$, hence is independent of $(y_i,r_i,\nabla_i)$, giving:
>
> $$
> \mathbb{E}\left[(1-\widehat\rho_{K-1}^{\text{loo},i}) \cdot r_i\nabla_i\right]=\mathbb{E}[1-\widehat\rho_{K-1}]\,\mathbb{E}[r\nabla].
> $$
>
> For the baseline term,
>
> $$
> \mathbb{E}\left[(1-\widehat\rho_{K-1}^{\mathrm{loo},i})\,b^{\mathrm{loo},i}\nabla_i\right]=\mathbb{E}\left[(1-\widehat\rho_{K-1}^{\mathrm{loo},i})\,b^{\mathrm{loo},i}\right]\mathbb{E}[\nabla_i]=0
> $$
>
> since $b^{\mathrm{loo},i}$ excludes $i$ and $\mathbb{E}[\nabla_i]=0$. Eq. (15) is then obtained by regrouping the per-sample LOO sum into correct/wrong groups (Eq. (13)), which does not affect unbiasedness.
>
> We added a short appendix subsection B.3 (“Proof of unbiasedness for RLOO$_K$ / Eq. (15)”) to make this argument explicit.

---

> ### Author Response · Authors · 2026-01-21
>
> **Claim 2 biased estimator**
>
> Correct. In the forward-engineering recipe (Sec. 5.2 and newly added Sec. 5.3), replacing population quantities by plug-in batch estimates does not in general preserve unbiasedness. The role of the forward-engineering recipe is to show how one can derive an advantage-shaping rule whose population update corresponds to the surrogate reward one starts with; it is not intended to guarantee unbiasedness at finite $N$. As one concrete example, Claim 2 recovers vanilla GRPO by forward-engineering the surrogate $F(\rho)=2\arcsin(\sqrt{\rho})$; as is well-known, GRPO itself is a biased estimator.
>
> We already note this explicitly in the main text. E.g. in Sec. 5.2 under paragraph “Biased versus unbiased estimation.”:
>     “In the derivation of Claim 2, contrary to our construction of RLOO$_K$ in Sec. 3, we did not insist on maintaining unbiasedness of the gradient estimator. $\frac{1-\hat\rho_K}{1-\hat\rho_K}$ is a biased estimator … and even replacing $\hat\rho_K$ with a leave-one-out estimate would not remove this bias since the product of two unbiased estimators is generally biased.”
>     “However, note that even the strong vanilla GRPO baseline does not implement an unbiased empirical estimate of the gradient.”
>
> Thus, the bias is a property of the advantage-shaping methods we are unifying (e.g. vanilla GRPO and Pass@K GRPO by Chen et al.), not a flaw in the analysis.
>
> **GRPO has KL, clipping**
>
> We have now made these assumptions explicit at the first introduction of GRPO in the main body (pg. 5) with a pointer to App. A, which states the full GRPO objective and discusses how it relates to our surrogate-reward view.
>
> **Actual training effectiveness:**
>
>
> Thank you for the suggestion. Our paper's primary contribution is conceptual: to interpret and unify existing Pass@K policy-gradient methods via a surrogate-reward lens, and to provide a general forward-engineering recipe for deriving advantage-shaping rules from a chosen surrogate *F(ρ)* (with or without regularization), or equivalently from effective gradient weights *ω(ρ)*. We also note that the main algorithms we analyze and connect (RLOO, GRPO, GRPO_K, GRPÕ_K, Skew-R) are methods whose original papers report empirical evaluations; our contribution is to clarify their relationships and the implicit surrogate objectives they optimize.
>
> That said, following your suggestion, we have added preliminary experiments in Appendix D that directly optimize Pass@K-oriented surrogates using algorithms derived from our framework (e.g., GRPO_K). The results provide initial evidence that methods such as GRPO_K and Skew-R are competitive with existing baselines such as GRPÕ_K and Pass@K-mixed. We have also added a paragraph ("Empirical evaluation and surrogate selection" in Sec. 7) providing initial guidelines on how to leverage the (advantage ↔ weights ↔ surrogate) equivalence for algorithm design, while acknowledging that a comprehensive empirical study—selecting among many plausible surrogates/regularizers, carefully tuning them, and evaluating across tasks and compute regimes—requires substantial additional effort and resources and is best addressed in a dedicated empirical follow-up.

---

> > ### Comment · Reviewer_Rab9 · 2026-01-30
> >
> > Thank you for your response and for running the additional experiments. The writing issues and the notation/citation formats are now resolved. Regarding the theoretical derivations, some of the less straightforward cases make more sense after reading your explanation. For Eq.15, in addition to Appendix B.3, it would be helpful to add a brief remark in the main text summarizing this point.
> >
> > I agree that the paper's main contribution is on the theoretical side, so a toy example with the 0.5B model that demonstrates the empirical effect at a high level should be sufficient. As I mentioned in my original review, I believe the community would benefit from these theoretical results, and they may inspire further empirical methods building on your work. The correctness and effectiveness in actual training will likely be tested by the community. I will continue reading the remaining details; if no further concerns arise, I will directly submit my recommendation for acceptance.

---

> > > ### Author Response · Authors · 2026-02-05
> > >
> > > Thank you for the prompt response and again for your careful reading and helpful feedback throughout the review process. We appreciate your support of our work.

---

### Review · Reviewer_QEqk · 2025-12-22

**Summary Of Contributions:**

This paper studies policy-gradient optimization for Pass@K objectives in reinforcement learning with verifiable rewards. It shows that direct Pass@K policy gradients and heuristic advantage-shaping methods (e.g., GRPO^K) can be unified through a surrogate reward perspective, where advantage shaping is asymptotically equivalent to optimizing smooth transformations of the Pass@K success probability. The authors further propose a forward-engineering framework for designing new advantage-shaping methods from surrogate rewards and interpret hard-example upweighting as reward-level regularization, with an entropy-augmented variant as an illustrative example.

**Additional Comments:**

**Strengths**: The paper provides a clear and technically sound unifying perspective on Pass@K optimization and advantage shaping in RLVR. The surrogate-reward viewpoint is elegant and helps reconcile several recent methods under a common framework.

**Weaknesses**:

1. In practice GRPO is usually run in a PPO-style way (old-policy rollouts, multiple epochs), but the surrogate-reward story is derived in a clean fully-online/on-policy setting.

2. Several of the paper's main correspondences rely on large-$N$ plug-in asymptotic approximations. In typical training, however, the group size is often small (e.g., $N \in \ {8,16\ } $), so the empirical success rate $\hat{\rho}$ can be noisy and the resulting reweighting can deviate from the asymptotic picture.

3. The forward-engineering recipe makes it easy to map a chosen surrogate $F$ to a reweighting rule via $F'(\rho)$, but the paper does not spell out practical design criteria for picking $F$ (e.g., avoiding overly sharp weights near $\rho\approx 0$ or $1$, or robustness to plug-in noise when $N$ is small). A short discussion of surrogate-selection trade-offs (stability vs.\ aggressiveness of reweighting, sensitivity to plug-in noise, behavior near $\rho\approx 0/1$) would make the recipe more actionable.

**Audience:**

Yes

**Audience Explanation:**

Yes — the results are directly relevant to researchers working on RLVR and LLM training.

**Claims And Evidence:**

Yes

**Claims Explanation:**

Yes — the paper’s claims are supported by clear and technically sound theoretical derivations, with assumptions and limitations explicitly stated.

**Requested Changes:**

1. It would help to add a short discussion of how the surrogate-reward view should be interpreted in the settings people actually use (e.g., small $N$, and GRPO implementations with clipping and multi-epoch updates).
2. (optional) Beyond the analytical weight-shape plots, a simple toy RLVR simulation showing how the different reweightings change the actual update behavior (and/or Pass@K trajectories) would make the paper more concrete.

---

> ### Author Response · Authors · 2026-01-21
>
> Thank you for your useful feedback. We also appreciate the encouraging comments. We have revised the paper and respond to your specific points below.
>
> **W1. the surrogate-reward story is derived in a clean fully-online/on-policy setting.**
>
> That is correct. Our surrogate-reward interpretation (reverse-engineering) is derived in the fully-online/on-policy regime where importance ratios are 1 and clipping is inactive. We already acknowledge this in Sec. 7 (“Clipping”): when GRPO is run with multiple epochs and clipping, the simplified gradient form we analyze should be viewed as an on-policy approximation. We have now made these assumptions explicit at the first introduction of GRPO in the main body (pg. 5).
> That said, the main output of the surrogate lens is a family of GRPO-like updates with modified advantage scores that can be used within the standard GRPO training loop exactly as vanilla GRPO is implemented.
>
> **W2. main correspondences rely on large-$N$. In typical training, however, the group size is often small (e.g., $N$)**
>
> We agree. Our reverse-engineering statements are population/on-policy interpretations, and the forward-engineering “recipe” uses plug-in batch estimates; when $N$ is small these estimates can be noisy and the induced reweighting can deviate from the asymptotic form. Capturing such finite-sample effects (bias/variance of the plug-in factors and their interaction with the empirical policy-gradient estimate) is more challenging and beyond the current scope.
>
> That said, we believe the surrogate-reward lens is still useful in the small-$N$ regime as an organizing principle: it identifies the *population* objective/gradient field being approximated, while finite $N$ can be viewed as introducing additional stochasticity (and potentially bias) in that approximation. In the revision we have added this in Practical Considerations (“Biased vs Unbiased”) explicitly highlighting the limitation and noting finite-$N$ analysis as an important direction for future work.
>
> **W3. Surrogate-selection criteria / practicality**
>
> Thank you for the remark. We believe this is largely addressed by the revision changes we made in response to your and Reviewer Wzcq's feedback.
>
> In the revision we added a standalone treatment (new Sec. 5.3) that states the surrogate-reward viewpoint in general form and details the **forward-engineering recipe** for designing new advantage-shaping updates. We also clarified the resulting update already upfront in Sec. 1.2, with practical deployment considerations reiterated in Sec. 7.
>
> **Design objects.** For designing new algorithms, the most convenient objects to work with are the **surrogate reward** $F(\rho_\theta)$ (equivalently its derivative $F'$) or the associated **weighting function**:
> $$
> \omega(u) := F'(u)\,u(1-u)
> $$
> since these directly specify how the update reweights correct vs. incorrect samples in the weighted-gradient form in Sec. 5.3. In particular, this makes explicit how “sharpness” of reweighting is governed by the behavior of $F'$ (and hence $\omega$) near the boundary $u \in \{0,1\}$.
>
> **Implementation view (one-line GRPO plug-in).** The same recipe yields an explicit GRPO-style advantage estimator:
> $$
> \\widehat G_F(\\theta)=\\frac{1}{N}\\sum_i A_i^F\\nabla_i,\\qquad
> A_i^F=F'(\\hat\\rho)\\cdot (r_i-\\hat\\rho)
> $$
> and therefore surrogate-driven advantage shaping is essentially a *one-line change* in existing GRPO code: if a baseline implementation already computes GRPO advantages $A_i^{\mathrm{GRPO}}$, then:
> $$
> A_i^F = F'(\hat\rho)\sqrt{\hat\rho(1-\hat\rho)}\\,A_i^{\mathrm{GRPO}}
> $$
> i.e., multiply the implemented GRPO “scores/advantages” by the prefactor $F'(\hat\rho)\sqrt{\hat\rho(1-\hat\rho)}$, and then apply the usual clipping/KL choices (Sec. 7).
>
> We hope these additions clarify how to choose and operationalize a surrogate.
>
> **Toy experiments**
> Thank you for the suggestion. In the revision, we added complementary initial experiments in Appendix D that directly optimize Pass@K-oriented surrogates using our derived algorithms (e.g., GRPO_K). The results provide initial evidence that methods such as GRPO_K and skew-R can be competitive with existing baselines such as \tilde GRPO_K and Pass@K-mixed. That said, we have also added an explicit paragraph ("Empirical evaluation and surrogate selection" in Sec.~7) acknowledging that a comprehensive empirical study of the design space—identifying robust surrogate families across tasks and compute regimes—is beyond the scope of this paper and an important direction for future work. We also provide initial guidelines on how to leverage the unifying
> $$\text{advantage} \leftrightarrow \text{weights} \leftrightarrow \text{surrogate}$$
> framework for such future algorithm design.

---

### Review · Reviewer_Wzcq · 2026-01-02

**Summary Of Contributions:**

**Summary**

This work proposes to unify the two views for optimising Pass@K objectives in Reinforcement Learning with Verifiable Rewards (RLVR) by showing that advantages shaping techniques derived from GRPO, and direct optimisation algorithms derived from REINFORCE, are trying to address the same problem. This unified view is built on the introduced surrogate rewards. Furthermore, this work propose to a way to derive both existing and new advantage-shaping methods from a chosen surrogate reward, and vice-versa.

**Strengths**

- (S1. readability): This work makes a lengthy introduction on concepts in RLVRs for readers not so familiar with the related subjects. This work is well motivated and is generally well written. I especially find Table 1 very helpful to understand the newly derived methods using this work's proposed framework.

- (S2. importance): Unifying the two views in related works can be a key step in the developing new algorithms in the future. The reasons behind where algorithms down-weight easier examples while over-weight harder ones when optimising for the pass@K metric can be generalised by the bidirectional framework introduced in this work.


**Weaknesses**

- (W1. Readability): the flip side of S1, this lengthly introduction sometimes may cause confusion of where the central contribution of this work is proven. I had to re-read quite a bit to find where the contributions listed in sections 1.2 are proven.

- (W2. Missing introduction): For the lengthy introductions on the REINFORCE-style direct optimisation techniques and advantage shaping techniques, the surrogate rewards were not properly defined. Some texts addressing how surrogate rewards are utilised can be included. Furthermore, how to use a chosen surrogate reward to derive new advantage-shaping techniques can be explained further in detail. Currently it is only briefly explained in Section 1.2.

- (W3. Effectiveness of derived algorithms): I think this is the main weakness of this work. Although this work is more about theoretical contribution, I think it would be good to have some experiments, for example accuracy on some toy tasks, when directly optimising for pass@k using the proposed methods (i.e. $GRPO_k$). I think this would significantly strengthen this work by showing the effectiveness of proposed bidirectional relationship. Basically, to show that newly derived algorithms can work.

- (W4. Guidance for future practitioners): I feel somewhat confused on how to utilise these claims going forward. Given the new bidirectional relationships, where should we start if we were to design new algorithms? To my understanding is to somehow choose a surrogate reward; it would be great if the authors can include some starting points on how to do so.

**Audience:**

Yes

**Audience Explanation:**

Researchers studying LLM agents especially RLVR would find this work relevant. This unified view and the introduced framework can be quite impactful in deriving new algorithms.

**Broader Impact Concerns:**

None as far as I am concerned.

**Claims And Evidence:**

Yes

**Claims Explanation:**

To my understanding these claims are well supported by theories introduced in relevant sections. One small caveat is that the new algorithms, introduced by the framework of this work, were not empirically verified but only theoretically.

**Requested Changes:**

- (R1. missing references): There are quite a few missing references in-between the manuscript, especially regarding proofs referring to the appendices. For example, proofs in Appendix B are not referred/mentioned at all in the main texts. This somewhat reduces readability.

- (R2. add description): Address W2 by adding some descriptions regarding surrogate rewards, and how these surrogate rewards are utilised/can be utilised. Currently, the only description is in Section 1.2, which can be explained more in detail.

- (R3. add toy experiments): Address W3 by adding some toy experiment to show efficacy (doesn't have to be too big an experiment) comparing derived direct pass@k optimisation versus related work (i.e. versus $\widetilde{GRPO_k}$), or the derived 0/1 reward optimisation (i.e. $\text{Skew-R}$) versus related works.

- (R4. guidelines): It would be great if the authors can address W4 by including some guidelines for practitioners.

- (R5. minor, clarity): The captions in the figures can be improved by adding some conclusion in the captions to facilitate faster understanding. Especially Figure 3.

- (R6. minor, clarity): I might have misunderstood, however, the main discussions (sections claim 5) and plots (figures 1, 2, 3) assumes N>>K. In the proofs, this assumption is not clearly written. Furthermore, it is unclear what specific number of N and K can be expected. It would be great if the authors can clarify in the texts.

---

> ### Author Response · Authors · 2026-01-21
>
> Thank you for your careful reading and detailed feedback. We much appreciate your constructive suggestions and have revised the paper accordingly, as explained below.
>
> **W1/R1. Where the contributions listed in Sec. 1.2 are?**
>
> Valid point. In the revision, we add a brief “Roadmap” paragraph at the end of Sec. 1 that explicitly maps each listed contribution to the corresponding sections.
>
> **W2/R2. How to use a chosen surrogate reward to derive new advantage-shaping techniques can be explained further in detail.**
>
> Thank you for this helpful remark—we agree. In particular, since the surrogate-reward viewpoint is more general than our specific unification of existing Pass@K estimators, it merits a clearer, standalone presentation beyond the discussion in Sec. 1.2 and the instantiation with respect to Pass@K reward and algorithms in Sec. 5.1 and 5.2.
>
> We have made two changes in the revision:
>
> 1. **New Sec. 5.3:** We added a dedicated subsection that generalizes the reverse-engineering claim (and includes its proof for completeness) and explains how a *chosen* differentiable surrogate reward $F(\\rho_\\theta)$ can be used to **forward-engineer** an advantage-shaping / GRPO-style update. Concretely, it yields an explicit advantage-form estimator $\\widehat G_F=\\frac{1}{N}\\sum_i A_i^F\\nabla_i$, where the surrogate shapes learning via the scalar reweighting $F'(\\hat\\rho)$.
>
> 2. **Clarification in Intro:** We also clarified this resulting policy-gradient update *upfront* in Sec. 1.2, so the reader sees immediately how surrogate rewards are utilized to derive advantage shaping.
>
> **W3/R3. Toy experiments**
>
> Thank you for the suggestion. In the revision, we added complementary initial experiments in Appendix D that directly optimize Pass@K-oriented surrogates using our derived algorithms (e.g., GRPO_K). The results provide initial evidence that methods such as GRPO_K and skew-R can be competitive with existing baselines such as \tilde GRPO_K and Pass@K-mixed. That said, we have also added an explicit paragraph ("Empirical evaluation and surrogate selection" in Sec.~7) acknowledging that a comprehensive empirical study of the design space—identifying robust surrogate families across tasks and compute regimes—is beyond the scope of this paper and an important direction for future work. We also provide initial guidelines on how to leverage the unifying
> $$\text{advantage} \leftrightarrow \text{weights} \leftrightarrow \text{surrogate}$$
> framework for such future algorithm design. Please see also response below.
>
>
> **W4/R4. I feel somewhat confused on how to utilize these claims going forward. Guidance for future practitioners.**
>
> As mentioned, in the revision we have added a standalone treatment (new Sec. 5.3) that states the surrogate-reward viewpoint in general form and details the forward-engineering recipe for designing new advantage-shaping updates. Additionally, we clarified the resulting update already upfront in Sec. 1.2 with practical deployment reiterated in Sec. 7.
>
> **Where to start (design view):** For designing new algorithms, the most convenient objects to work with are the **surrogate reward** $F(\\rho_\\theta)$ (equivalently its derivative $F'$) or the associated **weighting function**:
>
> $$
> \\omega(u) := F'(u)\\,u(1-u)
> $$
>
> since these directly specify how the update reweights correct vs. incorrect samples in the weighted-gradient form in Sec. 5.3.
>
> **What you get (implementation view) + one-line GRPO plug-in:** The same recipe yields an explicit GRPO-style advantage estimator:
>
> $$
> \\widehat G_F(\\theta)=\\frac{1}{N}\\sum_i A_i^F\\nabla_i,\\qquad
> A_i^F=F'(\\hat\\rho)\\cdot (r_i-\\hat\\rho)
> $$
>
> This makes surrogate-driven advantage shaping essentially a *one-line change* in existing GRPO code: if a baseline implementation already computes GRPO advantages $A_i^{\\mathrm{GRPO}}$, then:
>
> $$
> A_i^F = F'(\\hat\\rho)\\sqrt{\\hat\\rho(1-\\hat\\rho)}\\,A_i^{\\mathrm{GRPO}}
> $$
>
> i.e., multiply the implemented GRPO “scores/advantages” by the prefactor $F'(\\hat\\rho)\\sqrt{\\hat\\rho(1-\\hat\\rho)}$, and then apply the usual clipping/KL choices (Sec. 7).
>
> **Heuristic starting points:** Conceptually, $\\text{Advantage Shaping} \\Leftrightarrow \\text{Surrogate Reward}$ mirrors the supervised-learning duality between modifying gradients and modifying loss functions. This suggests importing well-studied robust/imbalance-aware loss-design principles from supervised learning as starting points (translated into choices of $F$ or $\\omega$), and then using the forward-engineering recipe to obtain the corresponding RLVR/GRPO-style update. We highlight this parallel in the Conclusion and point to Sec. 7 for practical considerations (finite-$N$ bias, clipping, KL regularization).
>
> **R5. Captions**
>
> Thank you for the suggestion. We agree and revised the captions to include a brief takeaway, in particular adding one to Figs. 1 and 3, where it was missing.

---

### Decision · Action_Editor_h1tg · 2026-02-20

**Recommendation:** Accept as is

**Audience:**

Yes

**Audience Explanation:**

There is clear empirical evidence of problems in the field of policy gradient methods such as GRPO and how they are used at scale. This paper offers an interesting novel perspective that addresses real challenges and that will interest the community.

**Claims And Evidence:**

Yes

**Claims Explanation:**

While this is mostly a theory paper, it is supported by proofs and some empirical evidence of the phenomena described.